# Evolution of surface deformation related to salt extraction-caused sinkholes in Solotvyno (Ukraine) revealed by Sentinel-1 radar interferometry

Eszter Szűcs[1], Sándor Gönczy[2], István Bozsó[1], László Bányai[1], Alexandru Szakacs[3], Csilla Szárnya[1], Viktor Wesztergom[1]

[1] CSFK Geodetic and Geophysical Institute, Sopron, 9400, Hungary
[2] Department of Geography and Tourism, Ferenc Rakóczi II Transcarpathian Hungarian College of Higher Education, Berehove, Transcarpathia, Ukraine
[3] Institute of Geodynamics, Romanian Academy, Bucharest, Romania; Sapientia Hungarian University of Transylvania

*Correspondence to*: Eszter Szűcs (Szucs.Eszter@csfk.mta.hu)

**Abstract.** Rocksalt has remarkable mechanical properties and a high economic importance, however, the strength of salt compared to other rocks, makes it a rather vulnerable material. Human activities could lead to acceleration of the dissolution of soluble rocksalt and collapse of subsurface caverns. Although sinkhole development can be considered a local geological disaster regarding the characteristic size of surface depressions the deformations can result in catastrophic events. In this study we report the spatiotemporal evolution of surface deformation in the Solotvyno salt mine area in Ukraine based on Sentinel-1 interferometric synthetic aperture radar measurements. Although the mining operations were finished in 2010 several sinkholes have been opened up since then. Our results show that despite the enormous risk management efforts the sinkholes continue to expand with a maximum line-of-sight deformation rate of 5 cm/yr. The deformation time series show a rather linear feature and unfortunately no slowdown of the processes can be recognized based on the investigated 4.5 year-long data set. We utilized both ascending and descending satellite passes to discriminate the horizontal and vertical deformations and our results revealed that vertical deformation is much more pronounced in the area. Analytical source modelling confirmed that the complex deformation pattern observed by Sentinel-1 radar interferometry has a direct connection to the former mining activity and is confined to the mining territory. With the 6-day repetition time of Sentinel-1 observations the evolution of surface changes can be detected in quasi real-time, which can facilitate disaster response and recovery.

keywords: Solotvyno salt mining, sinkholes, Sentinel-1 radar interferometry, surface deformation

# 1 Introduction

Large-volume rocksalt deposits formed in the Carpathian-Pannonian Region in its both internal (in the Transylvanian Basin) and external (along the outer margins of the Carpathian fold-and-thrust belt) parts during Badenian times, when those areas were communicating with each other. The Transylvanian Basin is a unique rocksalt storage, where salt layers of a few tens of meters thick were deposited, and later deformed by diapirism. Large volumes of salt migrated towards the margins of the basin due to the uneven loading of the overburden forming two basin margin-parallel belts of large-sized diapirs (Krézsek and Bally,

2006). Likewise, the salt layer deposited at the outer margin of the Carpathians, was deformed generating diapiric bodies known and exploited at many locations along the East Carpathians and South Carpathians in Ukraine and Romania. Solotvyno (Aknaszlatina, acc. to its local Hungarian name), located at the Ukrainian side of the northern East Carpathians, is one of them. The western and eastern salt lineaments along the Transylvanian Basin, as well as the salt bodies along the external East- and South Carpathians offered a valuable mineral resource for centuries in the past. Salt is still extracted industrially at some of

them. Nowadays, salt mines are again in the spotlight but for a different reason. Many of them, particularly the abandoned ones, pose a threat for populated residential areas, infrastructure and environment because the uncontrolled processes of suffusion and collapse of the old underground exploitation cavities (Deák et al. 2007, Móga et al., 2015, 2019, Zechneret al., 2019). One of the severely affected localities is Solotvyno, where collapse of subsurface caverns in the past resulted in dolines, temporally filled with brine and have a size of 150-230 m in diameter. Fig. 1 illustrates the most recent sinkhole developed in

the area in 2012.

Continuous monitoring of the evolution of such depressions and progression of sinkholes developed on top of old manmade excavations are essential to delineate unstable topographic surfaces and identify risk-prone areas in order to mitigate the threats. This requires high-resolution spatiotemporal observations (e.g. a dense network of measurement points) to follow and map the dynamics of the ongoing processes. Such a goal can be accomplished by using recently developed Earth observation techniques

(Elliott et al., 2016, Li et al., 2016).

[Please insert Fig. 1. near here]

The objective of this study is to assess the current state of the salt mine-related deformations in the Solotvyno area caused by

ground collapses in the past and to identify potential small-scale and dynamic surface variations related to dissolution cavities, which could result in further ground depressions and collapses of the abandoned salt-exploitation cavities in the future. Although the issue is well-known, no dedicated terrestrial monitoring network has been installed in the area yet. Therefore, Sentinel-1 satellite interferometry can be a unique opportunity to support the early identification of areas prone to sinkhole occurrence. The impact of salt dissolution related cavity collapse is not merely a local problem, but it could be amplified to a

regional-scale issue in the future owing to the proximity of the Tisa river, one of the main rivers in Central-Eastern Europe crossing several borders (Ukraine, Slovakia, Hungary, Romania, Serbia). As the water infiltration caused sinkhole

development propagates the boundary of the mining area, inhabited areas will be endangered. Some parts of the city have already been evacuated. However, besides the economic losses, the ecological impact of migrating brine into the underground freshwater system and eventually into the Tisa river can be catastrophic. Recognizing the environmental threat, the European Commission devoted considerable funds to support emergency preparedness, environmental protection, safety and security in the area. Pollution of the freshwater river with brine can have unforeseeable consequences similar to the environmental disaster caused by sequences of metal pollution originating from Romanian mining accidents at the beginning of 2000.

## 2 Geological background and salt mining activities in Solotvyno

80% of the Ukrainian Subcarpathian territory is a mountainous area, the remaining 20% shows a low-lying flat topography including two parts: the Chop-Munkachevo plain and the Marmarosh or Solotvyno basin extending southward in Romania across the state border (Chis and Kosinszki, 2011, Nakapelukh et al., 2017). The Solotvyno basin hosts most of the shallow subsurface salt domes, 16 of 19 in number, known in the Ukrainian Subcarpathians (Bosevs'ka, 2015). Starting from the town of Khust in the west, this 50 km long and 22 km wide elongated basin extends with diminishing width in the ESE direction. Its Ukrainian part is bounded by the Vihorlat-Gutin Neogene volcanic range to the west and south, and the Carpathian thrust-and-fold belt to the north and east (Fig. 2).

[Please insert Fig. 2. near here]

The Solotvyno salt diapir has a remarkable size (Bosevs'ka and Khrushchov, 2011). It has an elongated pear shape, oriented in the northwest - southeast direction, with a surface area of approximately 1 million $m^2$, (Fig. 3. a and c). The largest depth of the salt body is ca. 2 km, whereas its width varies between 200 and 800 m in the westernmost and easternmost parts, respectively. The salt diapir lies beneath a thick layer of gravel, embedded in sandstone at an average depth of 25-30 meters. The formation is covered with a thin, grey salt layer, the so-called pallag (layers of salt and clay) which functions as a waterproof layer, hindering the formation of natural salt karst. The earliest salt extractions were started by the method of dissolution, where the pallag layer was missing. As a result of the large-volume industrial mining, it was inevitable to cut through the sealing pallag layer which raised the possibility of flooding. Another factor, which makes the situation much more severe, is the closeness of the Tisa river. The river bed changed during time as the diapir emerged, and nowadays the river flows around the salt body from southern direction. The gradient of the river bed is rather high, about 15 m within a few kms (Fig. 3. b.). During the great floods of the river the water infiltrates towards the salt body and causes serious damages, where the waterproof layers are missing. Such events happened in 1998, 2001 and 2007 (Móga et al., 2015, 2017) contributing to the shutdown of the mines.

Exploitation of salt in Solotvyno has a long history, however, the industrial-scale production only began in the 18th century. Since the starting of the first mine in 1778, a total of 10 mines were operating in the area (Fig. 3. d., Tabl. 1.). The 8th (Lajos)

and the 9[th] mines were used for the longest time span, the others were operated for a relatively short period of time. At the end of the 1980s and beginning of 1990s a new, so-called 10[th] mine was developed but there was no exploitation here at all (Bosevs'ka and Khrushchov, 2011). The available total salt reserves are estimated at about 2 billion tons. In the beginning of the 1990s about 10% of the Ukrainian salt production was mined in Solotvyno, with an estimated yearly rate of 500 thousand tons. Prior to the industrial extraction of rocksalt, only small-scale karst formations were observed at the surface of the area affected by mining. The development of salt karst processes accelerated in such a way after the beginning of the large-volume industrial mining operations, that nowadays it is almost impossible to mine rocksalt at all.

[Please insert Fig. 3. near here]
[Please insert Tabl. 1. near here]

Environmental and economic problems of Solotvyno salt mines gained more and more public attention after 2000 and at the beginning of 2010 the city was categorized by authorities as a disaster prone area. Several studies were conducted in the past related to the salt operations, but these were limited to the investigation of the location, geology and formation conditions of the salt diapir and the possibilities of salt exploitation. Gaidin (2008) has already drawn attention to the problems of salt karst formation processes in 2008. He analyzed karst development thoroughly, presented the dissolution process of the pillars and the resulting stability change of mineshafts. Diakiv and Bilonizhka (2010) presented an overview of the geological outline of Solotvyno salt mine and in addition they described the stage of salt karst formation in 2010 in detail. They also highlighted the poor conditions of the drainage system around the cultivated areas with an emphasis that it could lead to the leakage of an increased amount of freshwater into the karst system. Diakiv (2012) has also investigated the salt karst formation in the area based on the studies of two mines. Bosevs'ka and Khrushchov (2011) discussed the possibilities of disaster response and mitigation. Meanwhile, several research teams have simultaneously started to examine the ongoing surface processes and resulting formations mainly by ground-based geomorphological mapping procedures or participatory GIS analysis respectively (Móga et al., 2015, 2017, Onencan et al., 2018).

**3 Material and methods**

Interferometric synthetic aperture radar (InSAR) is a remote sensing technique which operates with microwaves. The satellite emits electromagnetic radiation and detects the signal reflected from the surface. By exploiting the coherent phase difference of the time-separated SAR scenes information on the possible ground deformation can be retrieved. The great advantage of microwave radar interferometry is that it allows to monitor areas in all-weather conditions and to study surface deformations of natural or anthropogenic origin even in areas with difficult access. The ESA Copernicus Sentinel-1 mission is the first of its kind in the sense that it ensures coordinated global observations with unprecedentedly wide spatial coverage and with unrivaled measurement frequency (6 days for Europe and 12 days for other parts of the globe) and freely accessible for users (Sentinel-

1 User Handbook). Therefore, Sentinel-1 radar interferometry enables the detection of surface topographic changes from small-scale to high magnitude with a high density of measurements points, and due to the frequent repetition time of the satellite it allows to study the dynamics of numerous surface processes (Strozzi et al., 2013, Elliott et al., 2016).

Sentinel-1 is a twin constellation (Sentinel-1A was launched in 2014, B in 2016) of C-band (wavelength ~ 5.5 cm) satellites
separated by 180° in orbit. The main acquisition mode of Sentinel-1 over land is TOPS (Terrain Observation of Progressive Scans), which employs the generation of the wide swath, so-called IW mode (250 km) SAR products with medium spatial resolution (ca. 5 m x 20 m in range and azimuth directions, respectively). The narrow orbital tube of the satellites and the almost synchronized bursts of SAR scenes makes the IW products suitable for interferometric analysis.

### 3.1 Dataset

To assess the surface deformation caused by salt dissolution we used the available Sentinel-1 SAR collection covering the area of interest. The time period covered by Sentinel-1 is more than 4 years and it is sufficient to investigate the longer-term (i.e. multi-annual) behavior of surface deformation processes and to detect possible new surface developments.  Both ascending and descending data sets were utilized to facilitate the separation of total line-of-sight (LOS) deformation into east-west and vertical components which are more easily interpretable  as well as to better constrain the analytical modeling with combined
data tracks. Details of the Sentinel-1 SAR dataset used in this study is summarized in Table 2. Significant change in surface scattering properties (e.g. snow cover) results in low quality interferograms, therefore winter scenes with snow cover were excluded from the analysis.

[Please insert Table 2 near here]

### 3.2 Multi-temporal, multi-baseline InSAR analysis

The interferogram formation of Sentinel-1 IW SLC (Single Look Complex) images requires co-registration in azimuth direction of extreme accuracy due to the strong Doppler centroid variation within each burst (for more details see e.g. Yague-Martinez et al., 2016, Fattahi et al., 2017). For the precise co-registration of S1 scenes we followed the strategy described in Wegmüller et al.. (2016). We applied S1 precise orbit ephemerides and the 1' resolution SRTM surface model (Reuter et al.,
2007, Shuttle Radar Topography Mission 1 Arc-Second Global Digital Object Identifier number: /10.5066/F7PR7TFT) to consider the effect of terrain topography during a matching procedure of SAR images, the result of which were further refined using the Enhanced Spectral Diversity method (ESD) in the burst overlapping regions to reach a co-registration accuracy of an order of 1/1000 pixel. The ESD considers the double difference interferograms to determine the fine azimuth offset, therefore it is required to have coherent regions in the burst overlapping area. This requirement was met with several difficulties
as the investigated area is sparsely populated and lacks phase-stable natural scatterers. Therefore, we applied a cascade co-registration strategy selecting a so-called primary reference scene in the middle of the time series in the early spring period. Two secondary reference scenes were selected for each year, one in spring when the vegetation hasn't started to thrive, and

one in late autumn when only light vegetation covers the surface. These S1 scenes were co-registered directly to the primary reference scene and the rest of the scenes were co-registered to the primary reference using the nearest secondary reference scene in time for ESD estimation. Based on the co-registered stack, the interferograms can be calculated in a standard way (Simons and Rosen, 2007).

Time series analysis of interferograms requires coherent scatterers with quite stable geometric and electromagnetic properties over time. There are various techniques to select pixels, either dominated by a single scatterer or using averaged (multi-looked), noise-reduced distributed scatterers. These measurement points form the base of the time series analysis of differential interferograms either computed from a single-reference or a multi-reference stack (for more details on the comparison of different techniques see e.g. Crosetto et al. (2016), Osmanoğlu et al. (2016), and Manunta et al. (2019). To capture the high deformation rate of sequential depression and to maximize the coherence offered by the short spatial baselines and high revisit time of Sentinel-1 mission, we used a multi-baseline approach of interferogram formation (usually called Small Baseline Subset - SBAS, Berardino et al. (2002)). The interferograms were generated using the Gamma software (Wegmüller et al., 2016). We considered pairs of four consecutive SAR scenes to include redundancy in the interferogram network, which facilitates reduction of errors. We utilized both phase-stable single scatterers (PS) as well as distributed targets (DS), which ensures long-term coherence. The initial set of PS candidates was selected based on the high temporal stability of the backscattering as well as the low spectral diversity. For the DS scatterers we used multilooking with a factor of 5 x 1 (5 samples in range and 1 in azimuth) to increase signal to noise ratio but keeping in mind the spatial extent of the sinkholes. Distributed targets resulted in a 15 m x 15 m pixel size in the range and azimuth direction, which enables the detection of localized deformation caused by surface depression. The flat-earth phase and topographic phase were removed from the interferograms. In the multi-baseline approach interferograms were unwrapped in space first, finding the unambiguous phase values. The phase unwrapping was accomplished in an iterative way with quality control, keeping PS and DS pixels for the next step, which satisfy the phase model with reasonably small (< 1 rad) residuals. A two-dimensional phase model involving height corrections relative to the reference model (SRTM heights mapped to radar coordinates) and linear deformation rate was chosen. The residual phase consists of non-linear deformation phase, atmospheric propagation delay, error in the height correction estimates and other noise terms. The spatially correlated, low-frequency part of the residual phase was separated by spatial filtering from the residual phase, since unwrapping residual phase of point differential interferograms is much simpler than unwrapping the original point differential interferograms. The whole process was iterated starting from dividing the area into patches, where the linear phase model approximation was suitable. Using a multi-reference stack based on consecutive SAR scenes, the deformation phase can be kept as small as possible. With the constant refinement of the phase model, a single regression was applied on the whole area. The main output of the regression analysis was the unwrapped phase. The various phase terms were summed up and then the unwrapped phases were connected in time and inverted to deformations using a least squares approach minimizing the sum of the square weighted residual phases (Berardino et al., 2002, Wegmüller et al., 2016). The atmospheric phase and non-uniform deformation phase are present in the time series of unwrapped phases. To discriminate between the two, we identified areas with high deformation rate and excluded those phase values to estimate

atmospheric propagation delay. Atmospheric phases were determined as a combination of height dependent atmospheric delay plus the long-wavelength component of the SBAS inverted residual phase. We used a low-pass filter with a characteristic length of 5 km. Therefore, long-wavelength (> 5 km) non-linear deformation was mapped into atmospheric correction.

However, the area affected by subsidence is rather localized, so we can assume no long-wavelength non-uniform deformation.

### 3.3 Decomposition of surface deformation

The ascending and descending satellite passes offer the possibility to resolve the observed LOS deformations to vertical and horizontal components (Hanssen, 2001; Pepe and Calò, 2017; Fuhrmann and Garthwaite, 2019). The viewing geometry of the LOS vector is defined by the $\vartheta$ incidence angle, measured between surface normal and look vector, and the $\alpha$ satellite heading

$$d_{LOS} = [-\sin\vartheta\cos\alpha \quad \sin\vartheta\sin\alpha \quad \cos\vartheta] \begin{bmatrix} d_{EAST} \\ d_{NORTH} \\ d_{UP} \end{bmatrix} (1)$$

Since the satellite passes pole-to-pole and illuminates the surface in right angle to the path, the observed LOS deformation is the least sensitive to movements in the north-south direction. This component is usually neglected and the horizontal component of the movement is interpreted in terms of deformation only in the east-west direction. Based on the $\vartheta$ incidence angle of the reflecting point of the ground (Table 2.) the $d_{UP}$ vertical and the $d_{EAST}$ east-west component of the deformation

can be resolved (Fig. 4):

$$\begin{bmatrix} d_{LOS}^{ASC} \\ d_{LOS}^{DSC} \end{bmatrix} = \begin{bmatrix} \sin\vartheta_1 & \cos\vartheta_1 \\ -\sin\vartheta_2 & \cos\vartheta_2 \end{bmatrix} \begin{bmatrix} d_{EAST} \\ d_{UP} \end{bmatrix} (2)$$

[Please insert Fig. 4. near here]

### 3.4 Source modeling

We modeled the deformation observed by InSAR in order to better understand the mechanisms responsible for the sinkhole growth, and constrain the location and depth of underground cavities which can result in sinkhole collapse in the future. The cavity deflation was modeled using rectangular dislocation sources (Okada, 1992, Segall, 2010) within a homogeneous and isotropic elastic half-space. We used a rectangular pressurized crack model, since deformations are presumably related to the destruction of abandoned mines. Despite their simplicity and the inherited approximations, analytical formulations are

convenient to model and explain deformation patterns described by a few model parameters. The elasticity assumption implies that the half-space obeys Hooke's law, therefore displacements are considered infinitely small compared to the characteristic size of source dimensions (Lisowski, 2007). The observed gradual subsidence (see sec. 4.1) also supports the assumption of pure elastic deformation. We fit simple Okada rectangular dislocation models to the InSAR data using a grid-search method to estimate the initial model parameters. We made an exhaustive search for the best-fitting models using the misfit function

$$\delta = \left[\sum_{i=1}^{N} \sum_{j=1}^{M} \left(d_i - d_{i,m_j}\right)^2 / N\right]^{1/2}, (3)$$

where $N$ is the total number of measurement points, $M$ is the number of source models, $d_i$ is the observed cumulative surface deformation and $d_{i,m_j}$ is the modeled deformation from the jth source model projected onto the satellite LOS.

To refine the source parameters and estimate associated uncertainties we performed a Bayesian probabilistic inversion (Bagnardi and Hooper, 2018). We modified the open-source GBIS (Geodetic Bayesian Inversion Software, http://comet.nerc.ac.uk/gbis/) code to handle custom source models of multiple rectangular dislocations. We also jointly inverted the cumulative ascending and descending InSAR data to determine deformation source parameters, i.e. horizontal dimensions and horizontal coordinates of rectangular source, depth of dislocation, strike angle of horizontal edge with respect to the north and opening of model (related to volume change), for every model in a single run. Within a Bayesian inversion approach the characterization of posterior probability density functions (PDFs) of source model parameters are accomplished by taking into account uncertainties in the data. The optimal set of source parameters can be extracted from the posterior PDF by finding the maximum a posteriori probability solution. The PDFs of source model parameters are determined from the likelihood function of the residuals between the observations and the model prediction weighted with the inverse of the variance-covariance matrix of the observations. The Bayesian inversion approach requires the quantification of errors in the data, which are assumed to be multivariate Gaussian with zero mean and covariance matrix. For multiple independent data sets, the likelihood function can be formulated as the product of the likelihoods of the individual data sets. To increase the numerical efficiency, the GBIS inversion algorithm samples the posterior PDFs through a Markov chain Monte Carlo method, incorporating the Metropolis-Hastings algorithm, with automatic step size selection. For more details, we refer to Bagnardi and Hooper (2018).

## 4 Results

### 4.1 Observations

Time series analysis of Sentinel-1 interferograms reveals surface deformations beneath the city of Solotvyno and its surrounding. Based on the scattering properties of resolution elements a large number of pixels were identified ( >35,000 on a ca. 9 km x 9 km area (supplementary Figs. 1, and 2) which densely cover the populated areas. Coherent points were also detected in the direct vicinity of the existing sinkholes in low-vegetation fields.

Fig. 5. shows the linear LOS displacement rate determined from Sentinel-1 ascending data. The spatial pattern of the surface deformation computed from the ascending satellite pass data clearly shows the circular outline of the deforming area around the main sinkhole. Maximum deformation rate reaches almost 5 cm/yr (point 3 in Fig. 5.) and located at the south-southeast part of the central sinkhole. The detected surface deformation forms patterns. Moderate, but persistent surface displacement with a magnitude up to 2 cm/yr is found north of the central sinkhole in an urbanized area (around point 1). Larger deformation

rates occur south from the central sinkhole. Here, we identified two clusters of anomalous points, one patch located south-southeastwards from the central sinkhole with deformation rates varying between 4-5 cm/yr (around point 3). The other pattern is concentrated at the southwestern edge of the larger deforming area (point 2) with an average rate of 1-3 cm/yr. A localized deformation zone was found farther, ca. 2 km away, from the mining area in the west direction, on the left bank of the Tisza river. It's average rate is 1.2 cm/yr and it is located in the area with some small lakes (point 5).

[Please insert Fig. 5. near here]

The deformation rate is almost linear in the whole area, individual deformation time series for some selected points (mentioned in the text above) is given in Fig. 7. No significant sudden movement was detected. The time evolution of the LOS surface deformation is shown in Fig. 9. along a cross-section directed almost north-south (cross-section A-B). The path was selected to have the highest point density and a natural neighbor interpolation method was applied based on the points satisfying some distance criteria (< 50 m) around the location of the profile. Location of cross-sections was selected to get information in roughly perpendicular directions of the area, it was chosen by visual inspection of point distribution. Fig. 9. nicely shows the different magnitudes of the deformation rate between the northern and southern parts of the area, as well as the highly linear characteristics of surface evolution.

[Please insert Fig. 6. near here]

Regarding the descending Sentinel-1 interferogram time series, the average deformation rate shows a similar pattern compared to the ascending pass. This suggests that the deformation shows primarily vertical behavior. Results from the MT-InSAR analysis are shown in Fig. 6. in terms of average linear deformation rate. The movement north from the central sinkhole is much more pronounced and affects an extended area (see point 5). Deformation possibly related to landslide activity is captured on the hillside in the upper northern part of the figure, which was previously identified by Velasco et al. (2017). This movement along the gradient of the slope was not pronounced in the ascending satellite geometry. The growth of the disturbed area in time is illustrated in Fig. 10. on an east-west cross-section (marked by C-D) in the northern part of the investigated area. The motion is very linear in time, but shows an asymmetric shape in the east- west direction. The western part of the cross-section (the distance from point C is about 600 m) is much more similar to a classical subsidence bowl, whereas the eastern side (between 600 m and 1200 m) shows a sharp step (section from 600 m to 800 m). This part of the area, which moves like a solid block, is covered with buildings. Fig. 8. shows the individual time series of the points marked in Fig 6.

[Please insert Figs. 7. and 8. near here]

[Please insert Figs. 9. and 10. near here]

## 4.2 Decomposition of surface deformation

The different acquisition geometry results in dissimilar measurement points for the ascending and descending passes. To overcome this limitation and combine deformation rate results from ascending and descending passes, the data sets are usually interpolated to a common grid resulting in so-called pseudo measurement points (Ferretti, 2014). Besides the common spatial grid, a common zero reference point is required to make datasets obtained from different acquisition geometries comparable.

In our study both data sets were referred to the same reference point, which is located far enough from the deforming area in the east direction (marked with white cross in the supplementary Figs). As the time series analysis revealed that the surface deformation is quite linear, we combined the deformation rates and not the deformations directly. Results of LOS decomposition are given in Figs. 11. and 12. for the vertical and east-west (positive towards east) directions, respectively. Interpretation of vertical and horizontal deformations are more straightforward than explanation of LOS displacements and gives a coarse estimate on the ongoing processes. As Fig. 11. and 12 show, vertical deformation is more pronounced in the area, the vertical velocities are about twice in magnitude compared to the horizontal velocities. Although the most recent cavity collapse related to the abandoned mines occurred around 2012 in the northern (9th mine) and in the southern (8$^{th}$ mine with the twin dolinas) parts of the area subsidence is still in progress. The highest rates, ranging from 2.5-4.5 cm/yr, are recorded in the inner zone of the central sinkhole, whereas at the northern and southern edges of the deforming area deformation is still remarkable with a magnitude of 0.5 – 1.5 cm/yr. Regarding deformations in the east-west direction (Fig. 12.), the magnitude of the displacement is much smaller, the maximum rate being about 2 cm/yr. The complex surficial depression pattern suggests that the subsidence presumably originates from several individual sources or from the superposition of subsurface caverns. The northern part of the deforming area clearly shows a westward displacement, whereas its southern part shows displacement towards the east. The lack of InSAR observations impedes the retrieval the whole deformation signal, i.e. the alternating east-west pattern generated by a single subsidence void, therefore it can be only concluded that the deforming area is still actively growing nowadays.

[Please insert Figs. 11. & 12. here]

## 4.3 Comparison with previous results

A previous study conducted by TRE Altamira (Velasco et al., 2017) investigated the potential of radar interferometry to monitor surface deformations caused by mining operations in the Solotvyno area in a framework of a contract with the Hungarian National Directorate General for Disaster Management. An exhaustive study was conducted based on archive C-band SAR datasets as ERS (1997-2001), Envisat (2002-2010) and Sentinel-1 (2014-2016) as well as high resolution X-band Cosmo-SkyMed (2016-2017, 4 months) acquisitions utilizing the SqueeSAR algorithm (Ferretti, 2014) which combines high resolution PS points with coherent DS scatterers. Although the study covers almost 20 years, the number of images used from former missions is quite low, for archive data set: 29 (ERS) and 30 (Envisat) images were used compared to the contemporary Sentinel-1 mission with 44 SAR scenes. Archive data covers the investigated time period unevenly, therefore the resolution of the deformation time series may be inadequate which inherently raises the question of potential signal aliasing. The large collection of data from ESA's Sentinel-1 mission with frequent acquisitions guarantees to maintain the coherence in general and fosters the analysis of the dynamics of surface deformation.

Using the C-band satellites, Velasco et al. (2017) could not really identify phase stable points in the direct vicinity of the sinkholes. Deformation rates (maximum 25 mm/yr away from the satellite, on average) occur in north and south directions from the main sinkhole mostly in urbanized areas. The three C-band results are very consistent with each other both in

magnitude and pattern. Investigations based on the Cosmo-SkyMed data identified much more measurement points and gave almost uniform point distribution for the whole area. Several locations were detected in the inner zone around the main sinkhole with a cumulated magnitude of deformation as high as 40 mm for 4 months, which is equivalent to more than 10 cm/yr average deformation rate assuming a constant characteristic. It is quite interesting, however, that the pronounced deformation pattern located north-east of the main sinkhole in an industrial area, detected in all C-band datasets, was not identified at all. Although it should have an average value around 5-7 mm/yr, taking into consideration the short timespan of the investigated dataset, no measurement points were identified in the area of question. The study by Velasco et al. (2017) utilized satellite data only for descending pass compared to our investigations. For a meaningful comparison of the two studies we focused only on the datasets from C-band satellites. Velasco et al. (2017) concluded that there was a lack of coherent scatterers over the area around the sinkhole and deformation rate could be detected for inhabited areas only. These surface changes concentrated mostly to the north of the deforming area, also identified in this study, but we detected measurement points in thinly vegetated areas too. The longer time span of this study confirmed that in the area of interest deformation is still ongoing. For the displacement history curves (Figs. 7. and 8.) it can be concluded that uniform deformation model is adequate to interpret the results, no acceleration or slowing down trend can be identified.

### 4.4. Source modeling

The coarse estimation of model parameters was accomplished by forward modeling varying source model parameters on a predefined interval using the misfit function of Eq. (3). Unfortunately, no reliable information is available on the exact position, extension, orientation and depth of the mining underground. The estimated depth of underground mines varies between 50 m to 400 m, from the center to the perimeter of the mining area. The approximate location of the mines was estimated based on the available maps. The parameter space of the dislocation models was constrained based on the rough location, geometry and orientation of underground mines available on maps as well as the approximate depth of the salt layer. Lack of coherence, either due to change in ground cover or high rate of deformation, does not allow to retrieve the entire deformation pattern associated with sinkhole evolution. Therefore, cumulative deformations from ascending and descending satellite passes covering the same time period were utilized simultaneously to increase the reliability of source model parameter estimation. The lack of deformation signal around the center of the area of interest makes it difficult to identify the number of source models required to explain the subsidence pattern. Our results suggest a quad-source configuration of subsurface cavities. The best-fitting model parameters are summarized in Tabl. 3.

To refine the dislocation model parameters and estimate associated uncertainties we performed a Bayesian probabilistic inversion (Bagnardi and Hooper, 2018). It requires the error estimation of the data sets. Noise covariance of individual interferograms has been well studied, the main error sources are the noise caused by the temporally correlated phase decorrelation and the spatially correlated atmospheric phase delay. Since InSAR observations are inherently relative, the additive phase delays make the accuracy of measurements strongly dependent on the distance. There have been several

endeavors to provide an error analysis of time series InSAR output (see e.g. Agram and Simons, 2015; Cao et al., 2018 and references therein), however, we followed the method of Parizzi et al. (2020) and estimated the variance-covariance matrix of InSAR data sets experimentally. As Parizzi et al. (2020) points out, short time separated interferograms (supported by Sentinel-1 mission with multi-baseline analysis) are much more dominated by atmospheric propagation delay rather than phase variation due to deformation. After atmospheric phase correction the interferometric measurement error is practically the residual

atmospheric phase delay, as short time separated interferograms can be considered deformation-free. The mean variograms of the residual atmospheric phase shows a stationary behavior and can be approximated by a covariance function. Since both deformation and average velocity are related to the phase by a scale factor, the error estimates can be simply computed. We used an exponential covariance model fitted to the data to determine the variance-covariance matrix of deformation in the Bayesian inversion. For both the ascending and descending data sets similar models were obtained with a moderate range

values of 2.4 and 2.2 km for the ascending and descending datasets respectively.

Best-fit model parameters obtained from the forward modeling were utilized as starting values for the Bayesian parameter estimation. During the inversion the parameters were allowed to vary within reasonable limits taking into account the geological constraints and information of past mining activity. The optimal model parameters are summarized in Tabl. 3., coordinates are given in a local rectangular coordinate system. Our final model assumes four rectangular-shaped subsurface

cavities, developed in the salt layer. One source with a rectangular dislocation (model #1) of size 24.1 m × 64 m is located above the eastern edge of working panels of mine 9[th] at an estimated depth of 199.7 m. This mine was closed in 2008 due to water inrush. The moderate value of volume change suggests that this depression is an early stage of sinkhole development. The second source model (model #2) lies approximately 400 m southwest far from the first one and has a horizontal dimension of 63.5 m × 187.8 m, the required height change explaining the deformation pattern is -1.2 m. The elongated shape in roughly

north-south direction of the source model is in agreement with the subsurface mining activity. Between the main corridors of mines 9[th] and 10[th], long working panels were cut with varying length between a few tens to a few hundreds of meters. The third dislocation model (model #3) is located in the western periphery of the area affected by deformations. The model is roughly symmetric with a horizontal side length about 80 m and located at a depth of 273.1 m. There are several shallow mines there (numbered by 1 to 5 on Fig. 3. d.), established around the 18-19[th] century. These were completely destroyed as the

numerous, small-scale dolines filled with brine indicate on the surface. The source model parameters suggest that the inverse modeling tried to find a global solution for the observed subsidence pattern. However, a single source is unable to sufficiently explain the complex deformation pattern; a number of near-surface, small-scale voids, related to salt dissolution are needed as well. The fourth source model (model #4) is located beneath the working panels of mine 8[th], where heavy subsidence occurred around 2010, which resulted in the formation of the twin lakes. The depth of the model is about 296 m, the horizontal extension

of the model is 72.3 m × 82.1 m. The estimated opening equals to approximately an 18,000 $m^3$ volume change. Taking into account the horizontal extension of the existing surface depressions of the nearby twin lakes, 15,000 and 17,000 $m^2$ respectively, our modeling results seem reasonable. The question, whether a new doline will form and will merge with the

existing two in the future whether the boundary of the area affected by subsidence will expand toward the south, requires further observations with other tools besides radar interferometry.

## 6 Discussion

Although the detection of small-scale, episodic deformation related to sinkhole generation can be challenging, several studies demonstrated the potential of radar interferometry to characterize the post-collapse deformation of sinkholes (Baer et al., 2002, Galve et al., 2015, Kim et al., 2016, La Rosa et al., 2018), and there have been successful attempts to identify precursory deformations before the catastrophic collapse (Nof et al., 2013, Jones and Blom, 2014, Malinowska et al., 2019). In this study Sentinel-1 radar interferometry was utilized to investigate surface evolution in a salt karst environment, where land cover differs significantly from area to area, where sinkhole detection can be monitored almost semi-automatically. Several factors such as, the complex geological conditions (relatively large and shallow salt structure), the specific hydrological setting (wet and warm continental climate with increased precipitation rate in warm season, close proximity of a main river) and the mining conditions (damaged waterproof layer, mining levels at different depth, cracks in the rock masses) increase the susceptibility of sinkhole evolution. In this regard a single case study cannot serve to draw general conclusions on sinkhole generation mechanism, however, the experiences can be invaluable for making assessments and optimizing future monitoring methodology.

The Sentinel-1 radar interferometry results show that the detection of sinkhole evolution is feasible in the area, although it should be remarked that the point density of PS and DS scatterers is not sufficiently homogeneous. The complexity of the mechanism driving surface deformation as well as the inadequate sampling of the deformation pattern was revealed while explaining the subsidence pattern above the mining area by utilizing analytical modelling.

InSAR analysis showed that there is a continuous subsidence with a pronounced linear trend. Therefore, as a first-order approximation elastic modeling was utilized to explain the sinkhole formation. The quantitative analysis of the source modeling is shown on Figs. 13. and 14. in terms of the LOS deformation determined from the best-fitting quad-configuration source model (top), the Sentinel-1 cumulative LOS deformations (middle) as well as the difference between the observed and modeled values (bottom), for both the ascending and descending passes respectively. It can be assessed that the main features of the subsidence pattern on the northern and southern periphery are reasonably well captured by the source models. However, the modeled deformation on the western part of the area does not fit the observations, especially when compared to the ascending data set, where modeled deformation overestimates the observed ones. As it was pointed out earlier, under this area the salt layer upwells close to the surface and many small dropout dolines have already formed and a single source model cannot adequately explain the surface deformation pattern.

Apart from larger discrepancies at some individual points the modeled deformation pattern fits well to the observed ones. However, one has to keep in mind when evaluating the inversion performance, that it was not possible to properly sample the deformation pattern with InSAR, as only the margins of the area were mapped adequately. Due to the sparse InSAR observation

distribution in the middle of the area, we could not fit a proper source model there, which can be seen immediately when inspecting the modeled and observed deformations along selected profiles given on Fig. 15. (same profiles as cross-sections A-B and C-D). Regarding the ascending data set, the same north-south oriented profile was used to check the model fit to InSAR observations as shown Fig. 9. to check the subsidence evolution in time. In the northern part (starting from point A to appr. 200 m) of the cross-section the source models are capable to explain reasonably well the observed deformation. The

misfit of the modeled deformation is characterized by a standard deviation (std.) of $\pm$ 0.49 cm. However, for the second half of the investigated profile, between 200 and 800 m, the modeled deformation differs significantly from the observed ones. The reason for the large discrepancies in the middle of the cross-section comes from the fact, that it was not possible to find a proper source model based on the very scarce InSAR observations in the center of the area. On the southern edge of the cross-section (between 700 and 800 m) the applied single source model is not capable of resolving the observed deformation.

Probably the InSAR derived deformations reflect the effect of more than one subsurface cavern. Fig. 15. b. shows the observed and modelled deformations in a roughly east-west cross-section (C-D, for the location please check Fig. 10.). Modeled deformation shows a reasonably sufficient fit to the InSAR deformations. The misfit of the model is characterized by a $\pm$ 1.87 cm std. The magnitude of the observed deformation is adequately described by the model, however, the location of extremities is slightly miss-estimated. The profile crosses the area in the north, where model #1 and model #2 is located. The effect of the

two source models can be separated on the modeled deformations. Of course, the fine details revealed by InSAR observations cannot be reproduced by analytical modeling. Despite the simple formulas, analytical models can produce reasonable first-order results of the subsurface processes. Besides the above limitations it should be mentioned, that the possible interaction between the sources were not considered during the computation. As Pascal et al. (2013) pointed out, superposition of analytical models requires attention for adjacent models.

Despite the limitations of the analytical formulas the assumed elastic rheology fits well to the temporal behaviour of surface deformation, which implies that the cavity evolution is in its early stage. However, it should be mentioned that a more complex rheology can better constrain the precursory subsidence of sinkhole-prone areas as Baer. et al. (2018) demonstrated. The gradual surface subsidence suggested by the elastic deformation also agrees with the geological conditions of the area. Geomorphological investigations (Móga et. al, 2015) confirmed that the main driving mechanism of sinkhole formation in the

area is much more like the mechanism of the perfect suffosion of non-cohesive soils, than the sudden dropout of cohesive soils. The spatial distribution of observed surface deformations advocates that the subsidence is confined to the territory of the mining area, which implies that natural salt karst processes are initiated and accelerated by the anthropogenic intervention. As no deceleration of the ground movement was observed, it suggests that the dissolution of subsurface salt layers has become a self-sustaining geological process.

## 7 Conclusions

Salt mining operations in Solotvyno obviously demonstrate the severe and long-term consequences of a reckless industrial salt exploitation. In a sense, the mining area now has become a natural laboratory, where salt karst processes evolving much faster than in carbonate rock can be studied in great detail. Based on Sentinel-1 SAR interferometry we have demonstrated that significant surface deformation is still ongoing nowadays related to the former salt mining operations. We revealed a cumulative line-of-sight deformation up to nearly 15 cm in 4.5 years. It was also shown that the deformation neither has accelerated nor has decelerated during the investigated time period. Elastic inverse modeling of the observed deformations was utilized to constrain the geometry of the subsurface cavities. Although Sentinel-1 interferometry was capable to capture the deformation history of individual points, the modeling failed in some cases due to the insufficient InSAR point density, resulted in significant discrepancies between observed and modeled deformation. Despite the limitations of the study it was shown that precursory sinkhole formation may be detected by Sentinel-1 interferometry, which is a cost-effective way to obtain an overview of ground movements. Our result suggests that further steps to be done by local authorities in order to stabilize the mineshafts and improve the drainage system. Although several sinkholes have been opened in the last 20 years the geomorphological processes haven't ended yet, but there are hundreds of meters of intact mining holes that could pose a natural geohazard risk in the future.

Acknowledgement

Sentinel-1 data are copyrighted by the European Space Agency, and are additionally distributed by the Alaska Satellite Facility. We thank the Editor, Prof. Mahdi Motagh and two anonymous reviewers for their careful and thorough reviews.

Author Contributions: Conceptualization E. Sz., V. W.; methodology and SAR data analysis I. B., L. B., Cs. Sz.; geological and geographical background S. G., A. S., interpretation E. Sz., V. W.. Writing the original draft and making corrections during the review process: ALL.

Funding: E.Sz, I. B., L. B. and Cs. Sz. were supported by the National Excellence Program grant No. 2018-1.2.1-NKP-2018-00007.

Conflicts of Interest: The authors declare no conflict of interest.

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

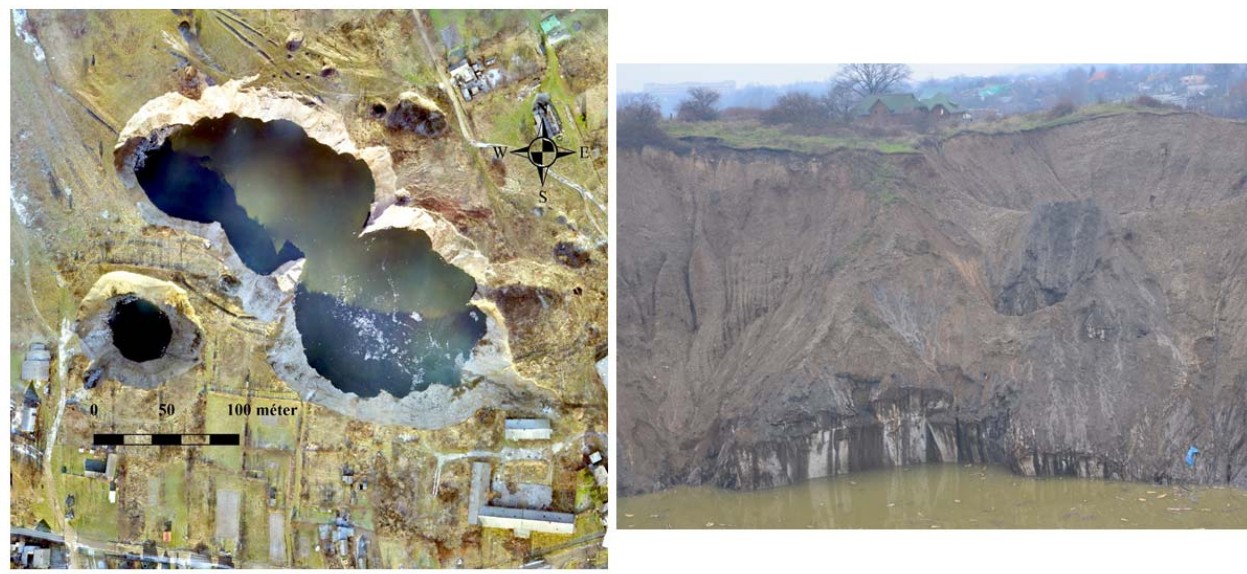

**Figure 1: Pictures of some sinkholes in the Solotvyno salt mining area (credit: S. Gönczy)**


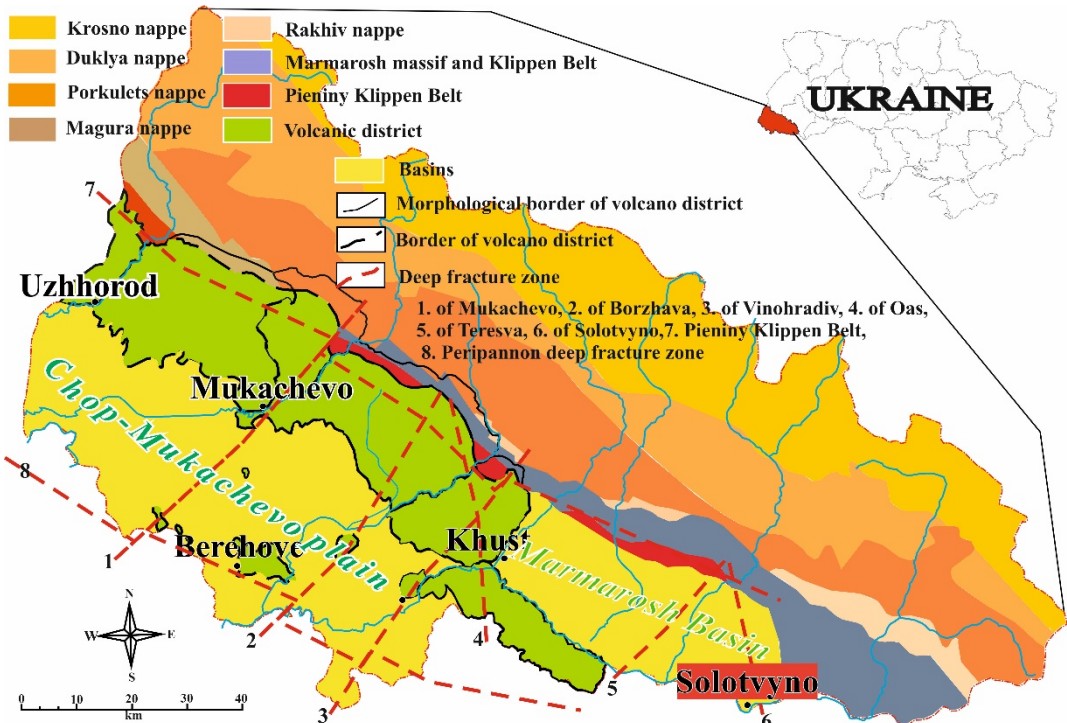

**Figure 2: Geological sketch map of the Ukrainian Subcarpathians compiled from © Cis, 1962; © Shakin, 1976; © Tyitov et al. 1979; © Herencsuk (ed.), 1981; © Glusko-Kruglov, 1986; © Kyiv State Cartographic Office 2000; © Kuzovenko (ed.), 2001**


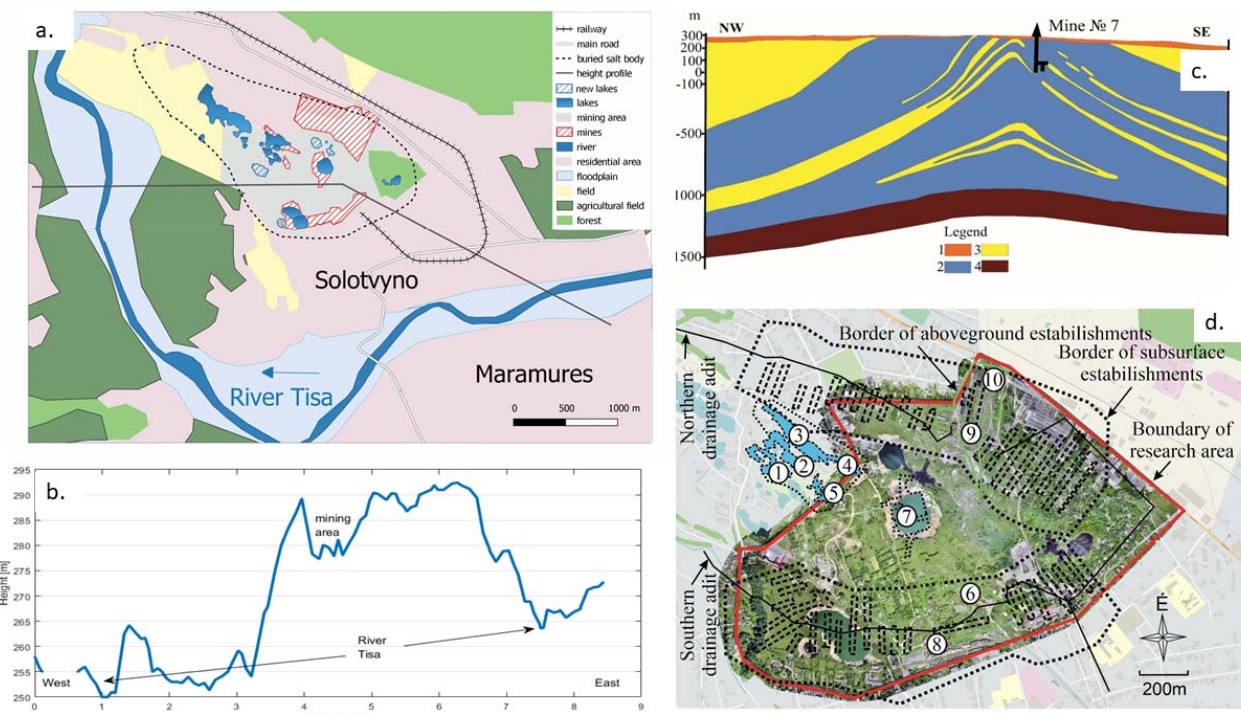

**Figure 3: The investigated area. | a. Overview map of Solotvyno and its surrounding, showing the approximate location of the salt dome. Lakes with hatch pattern show sinkholes opened after the detailed geomorphological mapping of © Móga et al. 2015. | b. Topographic profile across the section (black line) marked on figure a. | c. Geological cross section of the Solotvyno salt dome. Legend: 1. fluvial sediments; 2. Tereblya Formation; 3. Solotvyno Formation; 4. Novoselytsa Formation | d. Detailed picture of the area with the salt mines shown on a UAV map. 1. Kristina mine; 2. Albert mine; 3. Kunigunda mine; 4. Nicholas mine; 5. Joseph mine; 6. Old Louis mine; 7. Francis mine or mine 7[th]; 8. New Louis mine or mine 8[th]; 9[th] mine; 10[th] mine. Lakes are depicted in blueish shades.**



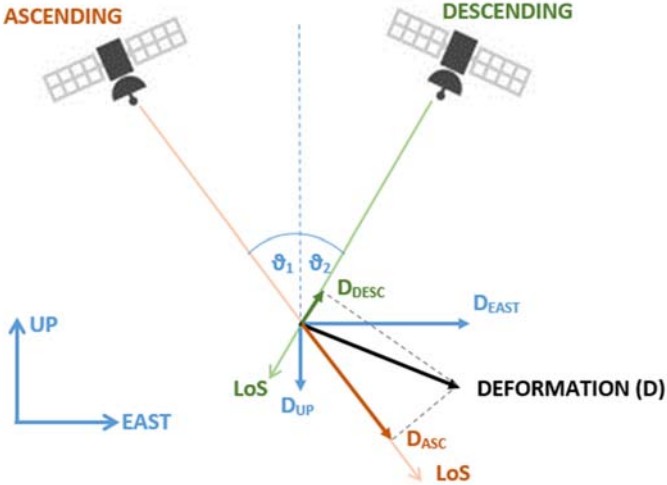

**Figure 4: Decomposition of the total deformation to ascending and descending LOS components as well as to vertical and quasi-horizontal (East-West) components**

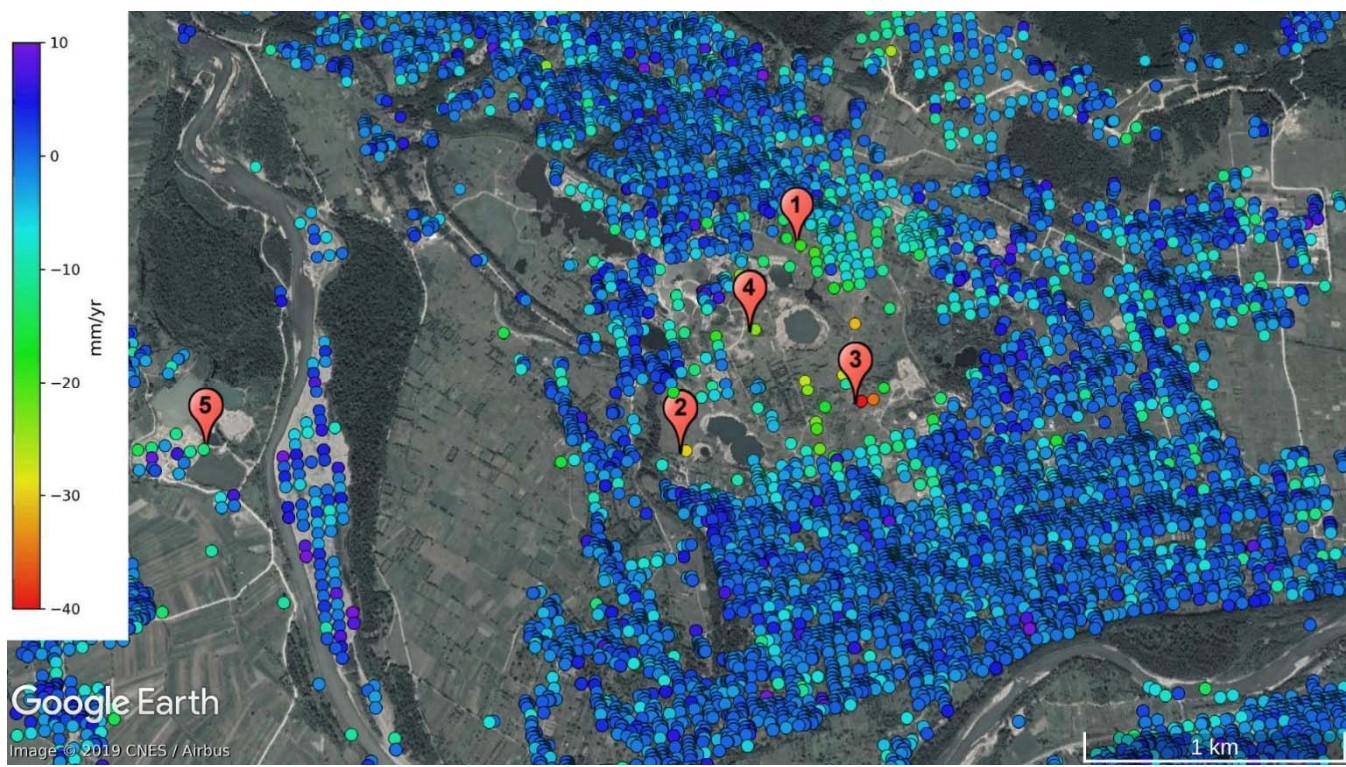


**Figure 5: Linear rate of line-of-sight deformation from Sentinel-1 ascending pass for the investigated area (see explanation of numbers in the text). Contains modified Copernicus Sentinel data [2014-2019], background © Google.**

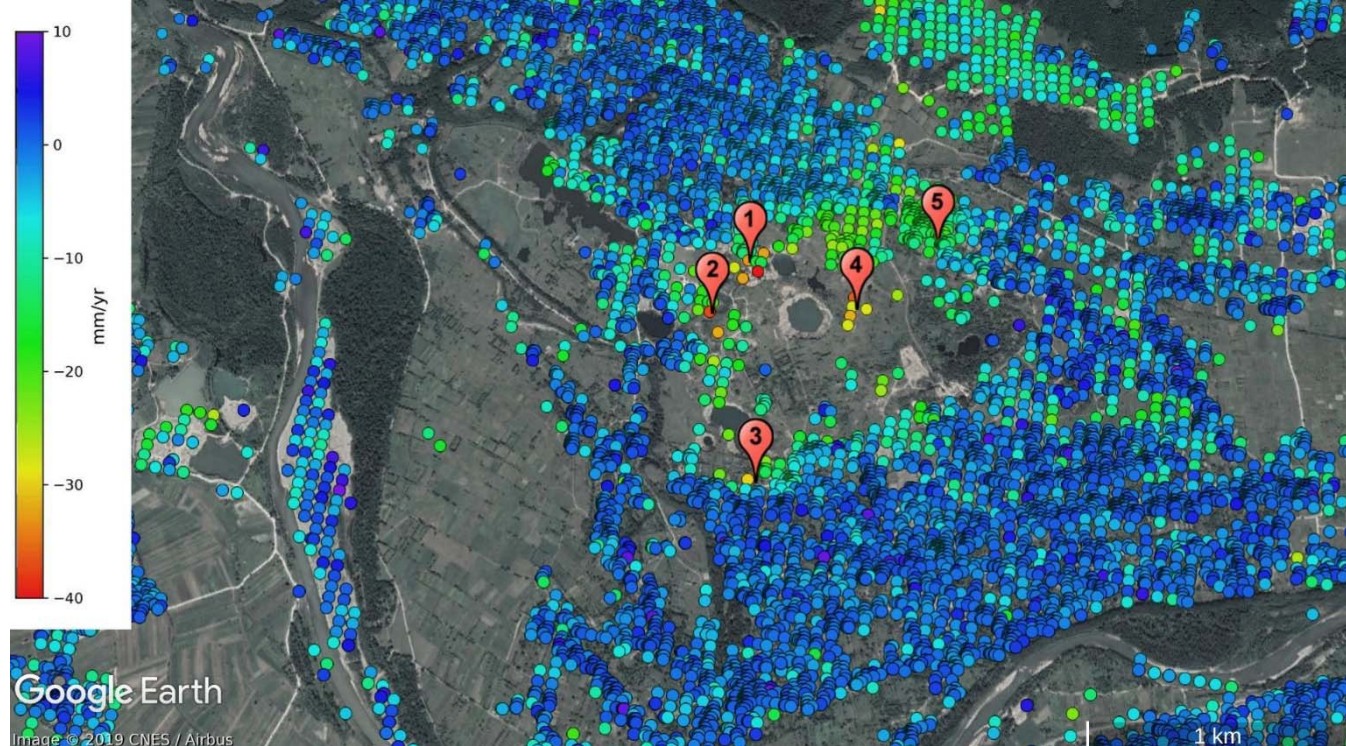

**Figure 6: Linearly estimated LOS deformation rate in mm/yr calculated from Sentinel-1 descending pass data for the investigated area. Contains modified Copernicus Sentinel data [2014-2019], background © Google.**

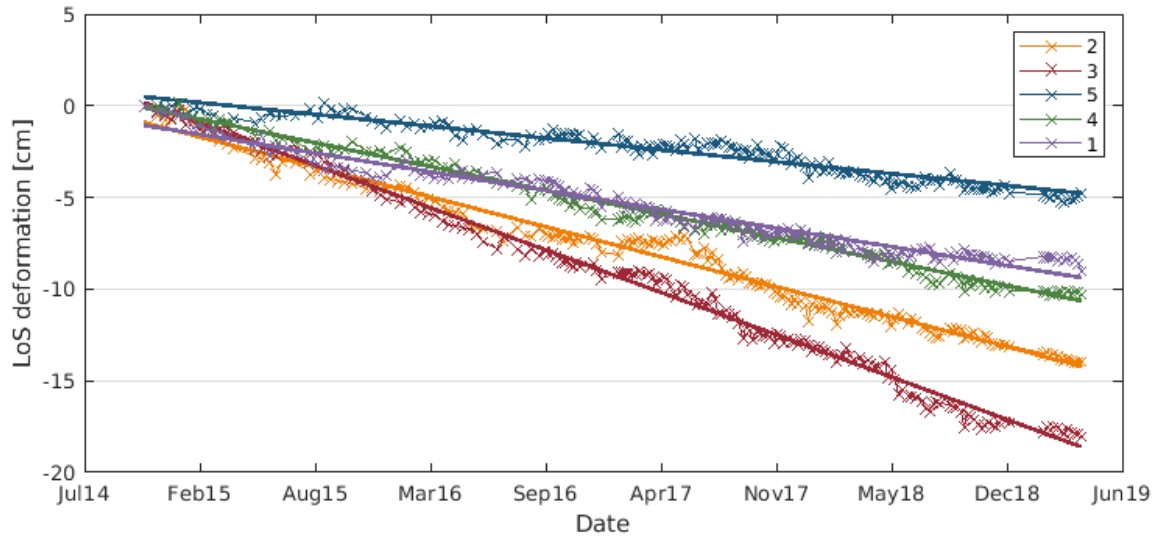

**Figure 7: LOS deformation of some selected points (see Fig 5. for location) determined from the ascending scenes**

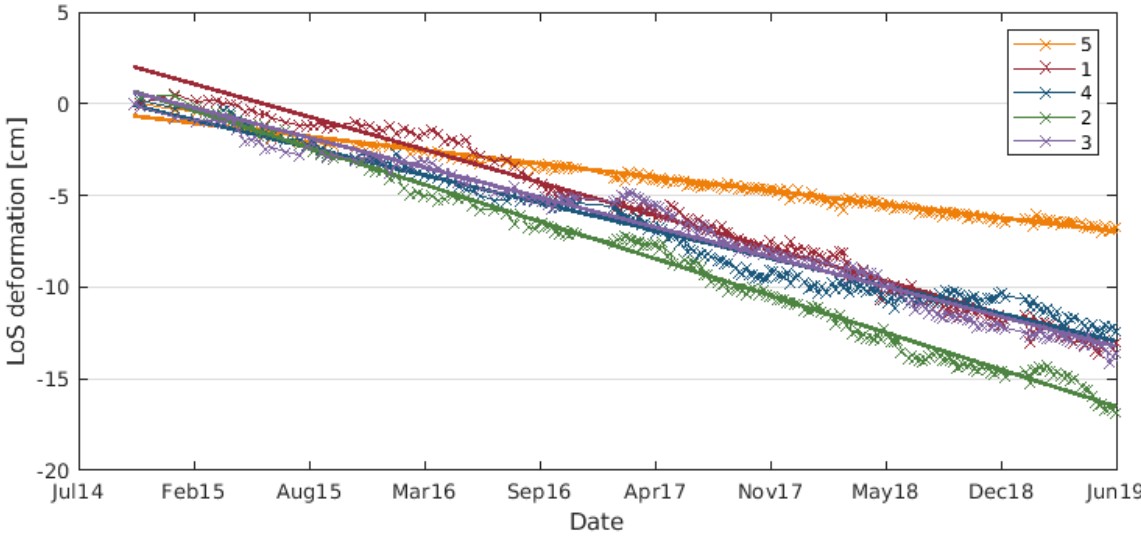


**Figure 8: Time series of some selected points (see Fig. 6. for location) from the analysis of Sentinel-1 descending scenes**

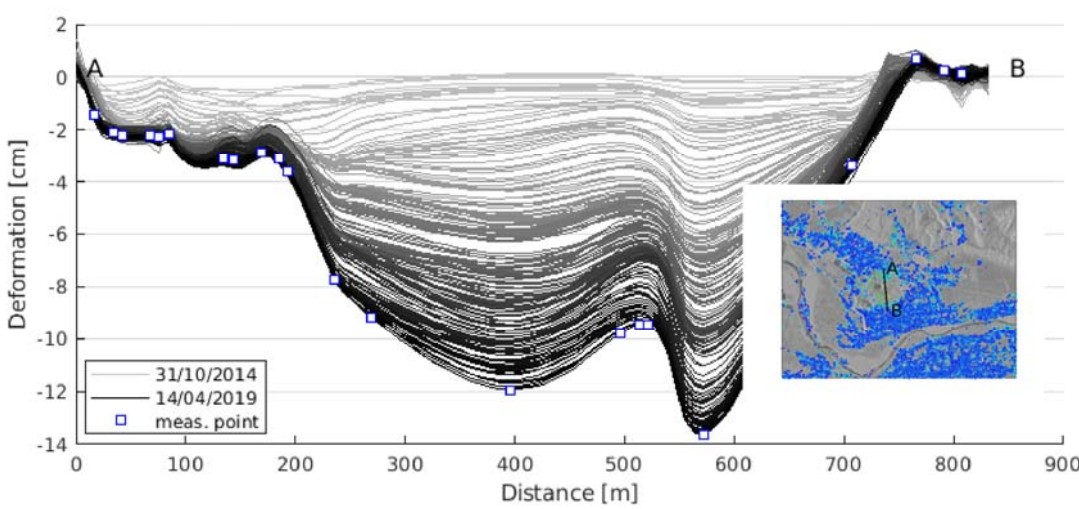

**Figure 9: Temporal evolution of LOS surface deformation for the AB cross-section from ascending Sentinel-1 data. For the location**
**of the section see the insert. Shade of grey varies from light to dark as time advances (lightest: October 2014, darkest: April 2019), blue squares show location of InSAR observations along the profile**

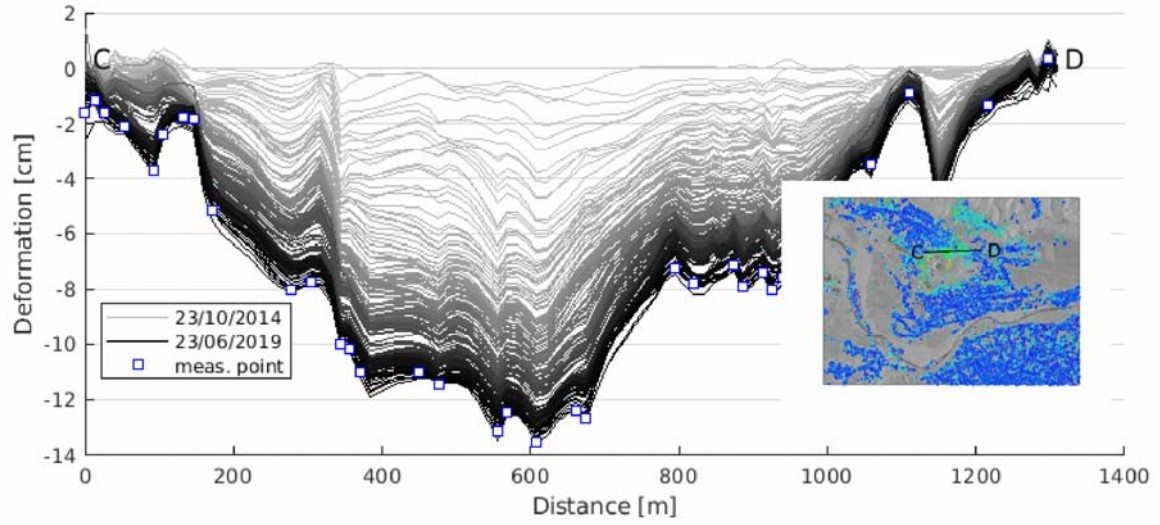

**Figure 10: Progress of LOS surface deformation from descending pass Sentinel-1 data. Location of cross-section is given in the insert. Shade of gray varying from light (October 2014) to dark (June 2016), blue squares show location of InSAR observations along the profile**

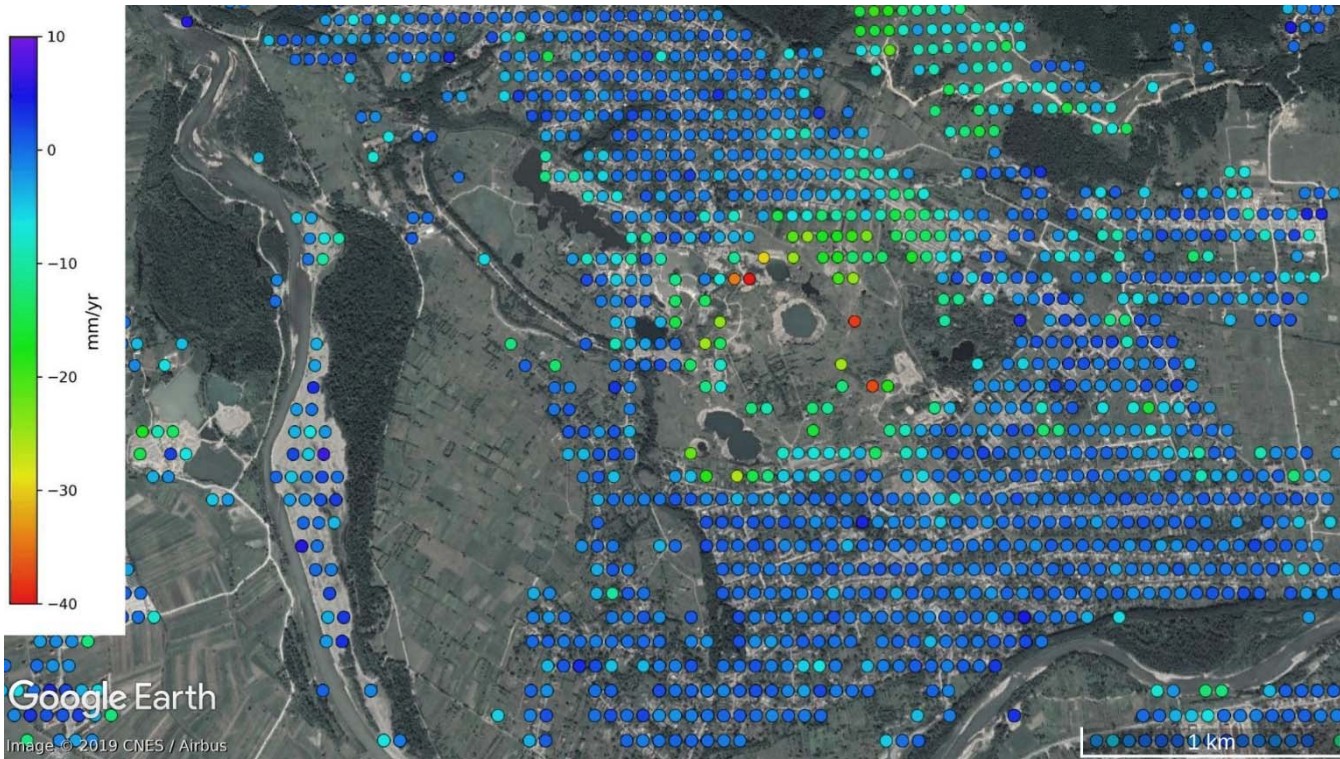

**Figure 11: Vertical deformation rate determined from Sentinel-1 data from track 29 and track 80. Contains modified Copernicus Sentinel data [2014-2019], background © Google.**


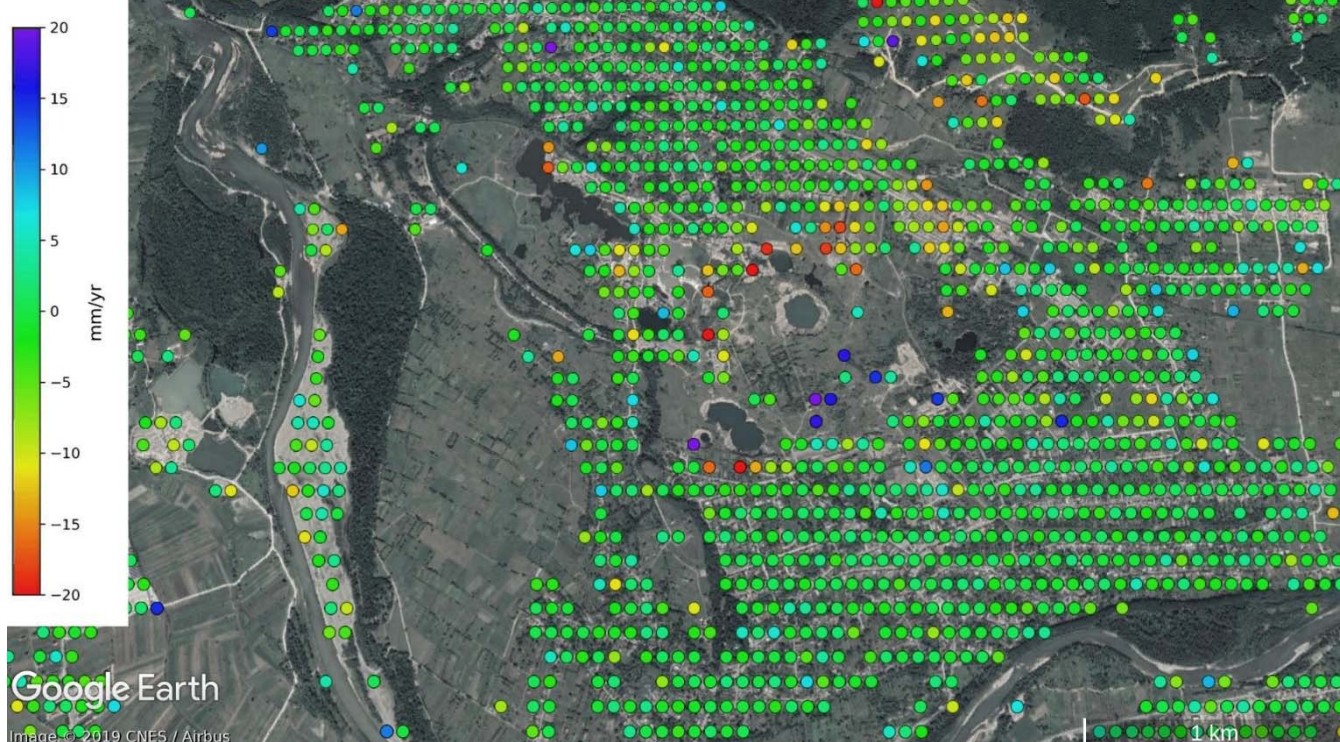

**Figure 12: East-west (positive in east direction) deformation rate of the area of interest computed from ascending and descending deformation rates. Contains modified Copernicus Sentinel data [2014-2019], background © Google.**

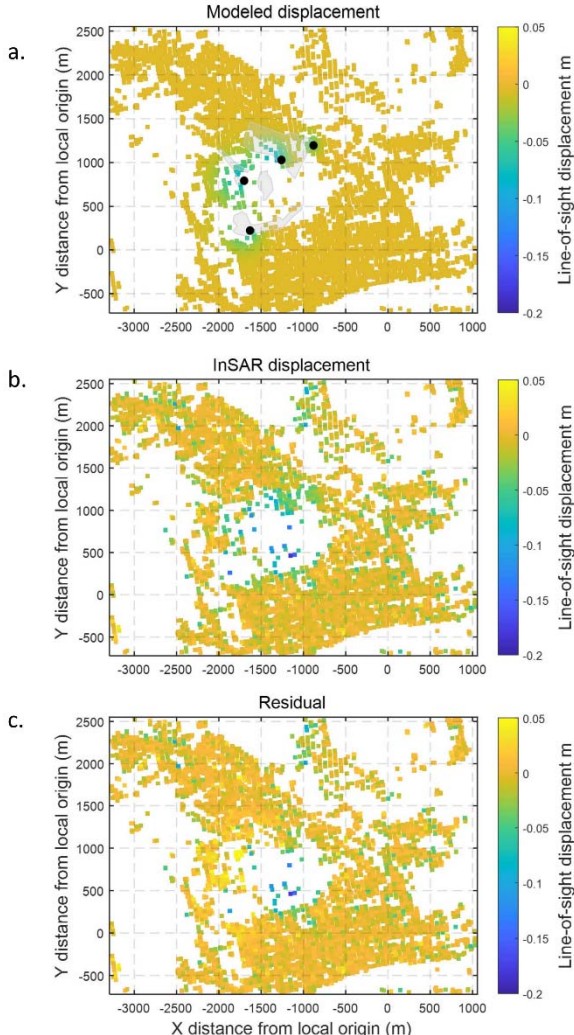

**Fig. 13. Cumulative LOS deformation a. computed from the quad-configuration source model (black dots denote the location of source models, grey polygons show the boundary of mines), b. deformation from the ascending Sentinel-1 observation and c. residuals after subtracting the best-fitting model**

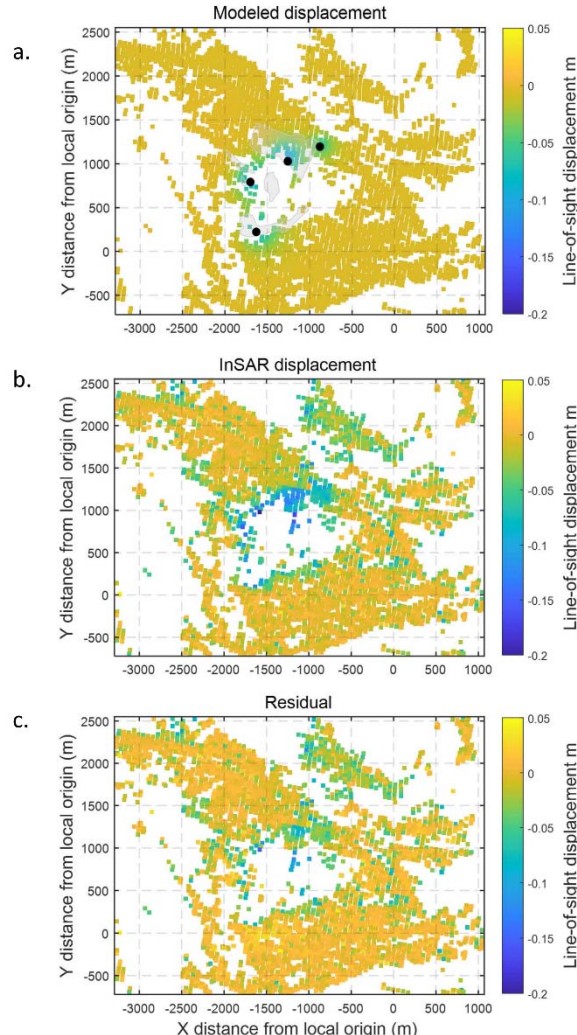

**Fig. 14. Cumulative LOS deformation a. of the best-fitting model using four dislocation sources black dots denote the location of source models, grey polygons show the mining area), b. deformation from the descending Sentinel-1 observation and c. residuals after subtracting the modeled displacement from the cumulative deformation**

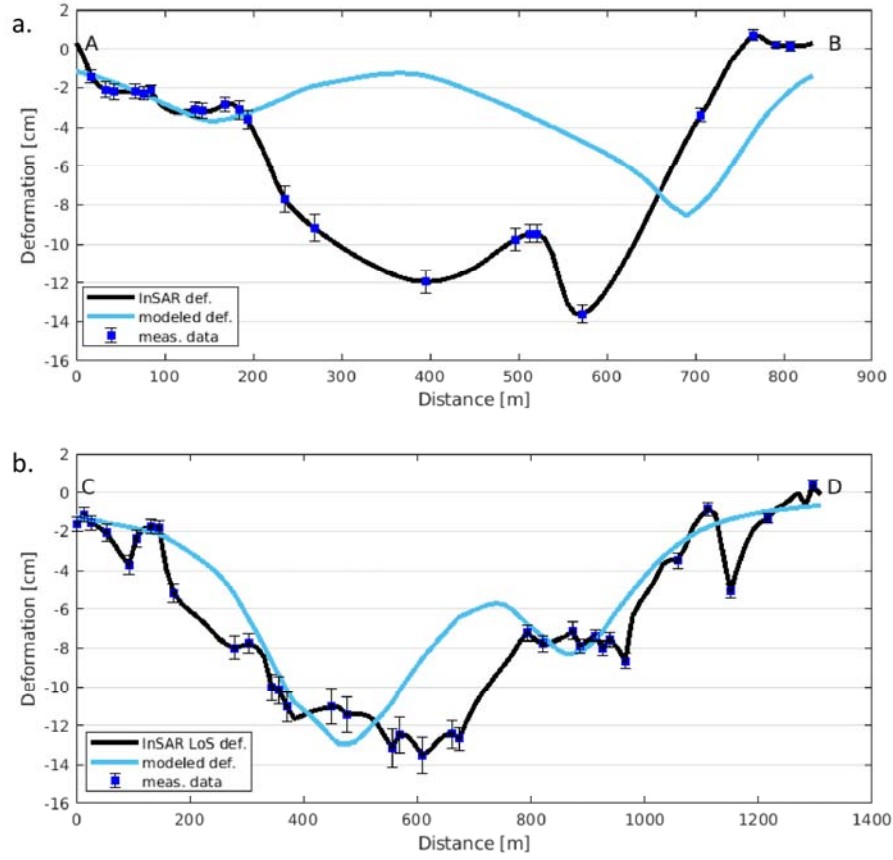

**Fig. 15. Observed cumulative LOS and best-fitting model LOS deformations along selected profiles (given on Fig. 9. and 10.) for the ascending (a.) and descending (b.) passes.**

| | The name of the mine | Start of extraction | Completion of extraction | Cause of the completion of extraction |
|---|---|---|---|---|
| 1 | Kristina mine | 1778 | 1781 | low quality salt |
| 2 | Albert mine | 1781 | 1789 | water inrush because the implosion of the surface |

| 3 | Kunigunda mine | 1789 | 1905 | water inrush because the implosion of the surface |
|---|---|---|---|---|
| 4 | Nicholas mine | 1789 | 1905 | water inrush because the implosion of the surface |
| 5 | Joseph mine | 1804 | 1850 | low quality salt, water inrush |
| 6 | Old Louis mine | 1804 | 1810 | low quality salt |
| 7 | Francis mine or mine 7th | 1809 | 1953 | water inrush |
| 8 | New Louis mine or mine 8th | 1886 | 2007 | water inrush |
| 9 | mine 9th | 1975 | 2008 | water inrush |
| 10 | mine 10th | It was made in the end of the eighties, but it has never worked | | |

**Table 1. Summary of mining activity in Solotvyno**

| Time span | Track | heading | incidence angle | Number of images |
|---|---|---|---|---|
| 20141031 - 20190414 | 29 (ascending) | -14.6° | 41.4° | 207 |
| 20141023- 20190623 | 80 (descending) | 194.6° | 36.3° | 211 |

**Table 2. Summary of Sentinel-1 data used in this study**

|  | model #1 | | model #2 | | model #3 | | model #4 | |
|---|---|---|---|---|---|---|---|---|
|  | I | R | I | R | I | R | I | R |
| length [m] | 36 | 24.1 | 72.5 | 63.5 | 80 | 78.2 | 80 | 72.3 |
| width [m] | 42 | 64.0 | 176 | 187.8 | 80 | 81.8 | 90 | 82.1 |
| depth [m] | 180 | 199.7 | 222 | 231.9 | 280 | 273.1 | 295 | 295.9 |

| strike angle [deg] | 5 | 12.8 | 21.5 | 19.1 | 22 | 18.7 | 21 | 17.1 |
|---|---|---|---|---|---|---|---|---|
| X center [m] | -870 | -880.3 | -1195 | -1259.1 | -1600 | -1630.8 | -1700 | -1700.5 |
| Y center [m] | 1160 | 1195.8 | 956 | 1029.6 | 230 | 224.9 | 810 | 793.1 |
| opening [m] | -3.2 | -2.8 | -1.4 | -1.2 | -3.9 | -2.3 | -3.8 | -3.1 |


**Table 3. Analytical model parameters used in the source modeling. "I" refers to initial, "R" to refined values; coordinates are given in local rectangular system, shown on Fig. 13.**