# Peer review of "Evolution of surface deformation related to salt extraction-caused sinkholes in Solotvyno (Ukraine) revealed by Sentinel-1 radar interferometry"

_Natural Hazards and Earth System Sciences, 2020_

## Referee Comment (RC1) · Anonymous Referee #1 · 23 Apr 2020

General comments: The paper uses InSAR to analyse surface deformation in an area of sinkholes formed due to salt mining in Solotvyno (Ukraine). The major results of the paper are: 1. two velocity maps and time series of LOS displacements, one in the ascending and the other in the descending Sentinel-1 tracks, for the years 2014-2019, with maximum LOS velocities of 5 cm/y. 2. Decomposition of the LOS displacements to vertical and E-W horizontal components. 3. Recognition of linear trends of deformation (no acceleration nor deceleration). The paper is local and mostly technical, showing some interesting results, however, it does not make any attempt to discuss

these results, their implications, or their contribution to our general understanding of sinkhole-related processes.

Specific comments 1. The geographical and geological background is far too long and detailed and is mostly irrelevant to the scope of the paper. The background sentences in the introduction are sufficient to understand the setting. 2. Materials and methods: lines 153-169 and 190-199 are introductory and background descriptions and should not appear in this section. Only lines 170-187 and 200-229 are relevant and should be combined with lines 53-62 to one section. 3. The authors present ascending and descending data and claim (line 254) that the "average (descending) deformation rate shows similar pattern as for the ascending pass. This suggests that the deformation consists mostly of vertical component.". This declaration has not been proved in any way, for example, by a graph comparing all ascending vs descending LOS velocities. Furthermore, similar patterns are not enough to prove that claim, the values should also be close (albeit moderated by the incidence angle). This should be shown. 4. The equation relating LOS to vertical and horizontal components should also include the heading angle between the track and the north. 5. The vertical velocities are about twice in magnitude (max ∼40 mm/y) compared to the horizontal velocities (max ∼20 mm/y). This means that deformation does not consist mostly of vertical deformation and that the horizontal movements should be considered and explained. Furthermore, Fig. 14 and lines 291-292 show that "The northern part of the deforming area clearly shows a westward displacement, whereas its southern part shows displacement towards the east". This means horizontal movements away from the subsidence centre (the sinkhole), which is counterintuitive, and should be explained and/or discussed. 6. Figures 10 shows a cross sections over an area that is highly incoherent, while the lines are continuous from side to side. The authors should explain how this section was made and make it clear where are the true points and where are the lines based on interpolation. 7. Line 183-184: "separation of total line-of-sight (LOS) deformation into east-west and vertical components which can help to understand the mechanism of sinkhole collapse and the progress of underground processes". The authors do not

show anywhere later in the paper any insight or discussion regarding the mechanism of sinkhole collapse and progress of underground processes. So what is the motivation for this separation to vertical and horizontal components? 8. Lines 257-259: can the authors please explain the large difference between the ascending and descending velocities in the "landslide" area? 9. Lines 261-264: The description of the two sides of the bowl is poorly supported by the figure. 10. Lines 305-306: "guaranties to maintain coherence" – coherence is definitely not maintained in the central area.

Technical corrections 1. The paper requires Language and grammar editing. Lots of sentences lack commas (,) to separate between parts of the sentence. Citation of previous studies should not be in brackets when they are the subject of the sentence. For example, line 143: (Gaidin, 2008) has already drew attention to. . ...., should be: Gaidin (2008) has already. . ... This type of error appears many times in the paper. 2. Line 115: change horizontal extension to areal extent 3. Line 256: what is MT-InSAR? Fig. 7 is like 6 but descending

---

## Referee Comment (RC2) · Anonymous Referee #2 · 21 May 2020

General comments: This manuscript applied InSAR to detect ground deformation related to salt extraction-caused sinkholes in Solotvyno (Ukraine). Both ascending and descending datasets from Sentinel-1 satellite were used to decompose horizontal and vertical displacement. Results found that the maximum LOS deformation is 5 cm/yr and the vertical deformation is much more dominant in the area. However, the aim of this paper is not completely clear: are the authors willing to prove the usefulness of In-SAR applied to sinkhole deformation (focus on the methodology) or are they interested on the ground deformations detected on the salt mines (focus on the case studies)?

[Figure]

Specific comments: 1. The authors talked a lot about geological settings in Section 2 "Geographical and geological background", but it looks not related to your discussions later in the paper. The same problem as in Section 3. Could you relate your deformation results to the geological settings and mining activities? Maybe you can add a section in discussion to talk about the relationship between deformation and geological setting/ mining activities. 2. In section 4 "Materials and methods", the first paragraph (line 155-175) is not related to this section, you have to focus on your SAR datasets and what method you developed/used to process your SAR data. My suggestion is to simplify your section 2 and discuss more about your methodology. Describe more about what software you used to process Sentinel-1 data, how do you deal with coherent pixel selection, or maybe how to mitigate atmospheric delay. 3. The description of decomposition method in Section 5.2 should move to the section "Methods". And in this section, you just need to discuss the decomposition results. 4. The equation of decomposition is wrong, the heading angle is missing. please refer to (Fuhrmann & Garthwaite, 2019). 5. I think your discussion is not enough, could you please talk about how the deformation results relate to the geological settings you described in Section 2.

Technical corrections: 1. Line 307, please check the citation format (Velasco et al., 2017) . And some of the same problems across the whole manuscript. 2. Line 221, Small Baseline Subset, SBAS -> Small Baseline Subset (SBAS) 3. Line 203, 1' resolution SRTM, is it 1 arc second?

---

## Author Comment (AC1) · 11 Jul 2020

We thank both anonymous reviewers for their constructive comments. We have addressed all of them in the following point-to-point rebuttal and we will incorporate the changes in the revised manuscript. As comments by the reviewers have some common remarks, we have sorted each reviewer's comments and grouped those that have common themes.

We marked our responses in blue, in detail (we also uploaded this document as sup-

plementary material, since we couldn't set font color, insert eq. and figures in the interactive comment).

**Reviewer 1, General comments: The paper uses InSAR to analyse surface deformation in an area of sinkholes formed due to salt mining in Solotvyno (Ukraine). The major results of the paper are: 1. two velocity maps and time series of LOS displacements, one in the ascending and the other in the descending Sentinel-1 tracks, for the years 2014-2019, with maximum LOS velocities of 5 cm/y. 2. Decomposition of the LOS displacements to vertical and E-W horizontal components. 3. Recognition of linear trends of deformation (no acceleration nor deceleration). The paper is local and mostly technical, showing some interesting results, however, it does not make any attempt to discuss these results, their implications, or their contribution to our general understanding of sinkhole-related processes.**

**Reviewer 2, General comments: This manuscript applied InSAR to detect ground deformation related to salt extraction-caused sinkholes in Solotvyno (Ukraine). Both ascending and descending datasets from Sentinel-1 satellite were used to decompose horizontal and vertical displacement. Results found that the maximum LOS deformation is 5 cm/yr and the vertical deformation is much more dominant in the area. However, the aim of this paper is not completely clear: are the authors willing to prove the usefulness of InSAR applied to sinkhole deformation (focus on the methodology) or are they interested on the ground deformations detected on the salt mines (focus on the case studies)?**

We agree with the reviewers for pointing out that the motivation of the paper is not fully clear. In the past decades satellite radar interferometry has become a widespread tool to detect subtle surface changes, like ground subsidence associated with sinkhole generation. The numerous studies available for the Dead Sea region clearly confirms that. With the advent of coordinated Earth observation, the near real time mapping of surface deformation processes become available. The recent paper by Nof et al. (2019) describes a semi-automatic early warning system that detects precursory subsidence

before sinkhole collapse primary based on SAR dataset.

Our study area is not investigated as thoroughly as the Dead Sea region, but it is also severely affected by sudden sinkhole collapses. Collapse of subsurface caverns in the past resulted in dolines, temporally filled with brine and have a size of 150-230 m in diameter. Although today it can be handled as a local problem, it will definitely become a serious issue in the future as the water infiltration caused sinkhole development will propagate through the boundary of the mining area and endanger inhabited areas. Some parts of the city have already been evacuated. Besides the economic losses, the ecologic impact of migrating salt water into underground fresh water system can be catastrophic, which can lead to a regional problem. Recognizing the situation, the European Commission devoted considerable funds to support risk reduction in the area.

The latest sinkhole collapse happened before the launch of the Sentinel-1 mission. Although the issue is well-known, no dedicated terrestrial monitoring network has been installed yet. Therefore, Sentinel-1 satellite interferometry seems to be the only opportunity to support the early identification of areas prone to sinkhole occurrence.

We also agree with the reviewers that a thorough discussion of the results is needed to improve the paper, providing more insight into the mechanisms responsible for sinkhole growth in the area. Although the observed deformation pattern is sparse (Reviewer 1 also noted this regarding the cumulative displacement profiles in comment 6) we tried to perform an inversion to get some information about the dislocation sources. This investigation will be incorporated into the revised version of the ms.

[revised manuscript text omitted]
 a one subsurface cavern. The bottom figure of Fig. 3. shows the observed and modelled deformations on a roughly east-westward cross section (the location of the section is the same as for Fig. 11. in the original ms.). Modeled deformation shows a reasonably sufficient fit to the InSAR deformations. The misfit of the model is characterized by a $\pm$ 1.87 cm std. The magnitude of the observed deformation is adequately described by the model, however, the location of extremities is a slightly miss-estimated. The profile crosses the area in the North, where model #1 and model #2 is located. The effect of the two source models

can be separated on the modeled deformations. Of course, the fine details revealed by InSAR observations cannot be reproduced by analytical modeling. Despite of the simple formulas, analytical models can produce reasonable first-order results of the subsurface processes. However, the possible interaction between the sources were not considered. As Pascal et al. (2013) points out, superposition of analytical models requires attention for adjacent models.

########## # Reviewer 1, specific comment 1. The geographical and geological background is far too long and detailed and is mostly irrelevant to the scope of the paper. The background sentences in the introduction are sufficient to understand the setting.

**Reviewer 2, specific comment 1. The authors talked a lot about geological settings in Section 2 "Geographical and geological background", but it looks not related to your discussions later in the paper. The same problem as in Section 3. Could you relate your deformation results to the geological settings and mining activities? Maybe you can add a section in discussion to talk about the relationship between deformation and geological setting/ mining activities.**

We agree with the reviewers that the given geological background is too lengthy in the present form. Our aim was to summarize the information available for the region, since most of the papers, textbook and maps are difficult to acquire and mostly written using the Cyrillic alphabet. We will shorten this section in the revised version of the ms. and add the results of the analytical source modeling to link the surface processes with geology and past mining activity.

########## # Reviewer 1, specific comment 2. Materials and methods: lines 153-169 and 190-199 are introductory and background descriptions and should not appear in this section. Only lines 170-187 and 200-229 are relevant and should be combined with lines 53-62 to one section.

**Reviewer 2, specific comment 2. In section 4 "Materials and methods", the first paragraph (line 155-175) is not related to this section, you have to focus on your SAR**

datasets and what method you developed/used to process your SAR data. My suggestion is to simplify your section 2 and discuss more about your methodology. Describe more about what software you used to process Sentinel-1 data, how do you deal with coherent pixel selection, or maybe how to mitigate atmospheric delay.

We thank the referees this remark. In the revised ms. we will merge the suggested paragraphs for the introduction to Material and methods.

We also will include more details on Sentinel-1 processing in Section 4.2 as follows (from line 221., original text given in black, please check supplementary file).

The interferograms were generated using the Gamma software (Wegmüller et al., 2016). We considered pairs of four consecutive SAR scenes to include redundancy in the interferogram network, which facilitates reduction of errors. We utilized both phase-stable single scatterers (PS) as well as distributed targets (DS), which ensures long-term coherence. The initial set of PS candidates was selected based on the high temporal stability of the backscattering as well as the low spectral diversity. For the DS scatterers we used multilooking with a factor of 5 x 1 (5 samples in range and 1 in azimuth) to increase signal to noise ratio but keeping in mind the spatial extent of the sinkholes. Distributed targets resulted in a 15 m x 15 m pixel size in the slant range, which enables to detect localized deformation caused by surface depression. The flat-earth phase and topographic phase were removed from the interferograms. In the multi-baseline approach interferograms were unwrapped in space first, finding the unambiguous phase values. The phase unwrapping was accomplished in an iterative way with quality control, keeping those PS and DS pixels for the next step, which satisfy the phase model with reasonably small (< 1 rad) residuals. A two-dimensional phase model involving height corrections relative to the reference model (SRTM heights mapped to radar coordinates) and linear deformation rate was chosen. The residual phase consists of non-linear deformation phase, atmospheric propagation delay, error in the height correction estimates and other noise terms. The spatially correlated, low-frequency part of the residual phase was separated by spatial filtering from the residual phase, since unwrapping residual phase of point differential interferograms is much simpler than unwrapping the original point differential interferograms. The whole process was iterated starting from dividing the area into patches, where the linear phase model approximation was suitable. Using a multi-reference stack based on consecutive SAR scenes, the deformation phase can be kept as small as possible. With the constant refinement of the phase model a single regression was applied on the whole area. The main output of the regression analysis was the unwrapped phase. The various phase terms were summed up and then the unwrapped phases were connected in time and inverted to deformations using a least squares approach minimizing the sum of the square weighted residual phases (Berardino, Fornaro, Lanari, & Sansosti, 2002; Wegmüller et al., 2016). The atmospheric phase and non-uniform deformation phase are present in the time series of unwrapped phases. To discriminate the two, we identified highly deforming areas and excluded those phase values to estimate atmospheric propagation delay. Atmospheric phases were determined as a combination of height dependent atmospheric delay plus the long-wavelength component of the SBAS inverted residual phase. We used a low-pass filter with characteristic length of 5 km. Long-wavelength (> 5 km) non-linear deformation was mapped into atmospheric correction. However, the area affected by subsidence is rather compact so we expect no long-wavelength non-uniform motion.

**Reviewer 2, specific comment 3. The description of decomposition method in Section 5.2 should move to the section "Methods". And in this section, you just need to discuss the decomposition results.**

Thank you for the suggestion, we will move the paragraph describing LoS decomposition to Sec. 4. in the revised version of the ms.

########## # Reviewer 1, specific comment 3. The authors present ascending and descending data and claim (line 254) that the "average (descending) deformation rate shows similar pattern as for the ascending pass. This suggests that the deformation consists mostly of vertical component.". This declaration has not been proved in any

way, for example, by a graph comparing all ascending vs descending LOS velocities. Furthermore, similar patterns are not enough to prove that claim, the values should also be close (albeit moderated by the incidence angle). This should be shown.

**Reviewer 1, specific comment 5. The vertical velocities are about twice in magnitude (max 40 mm/y) compared to the horizontal velocities (max 20 mm/y). This means that deformation does not consist mostly of vertical deformation and that the horizontal movements should be considered and explained. Furthermore, Fig. 14 and lines 291-292 show that "The northern part of the deforming area clearly shows a westward displacement, whereas its southern part shows displacement towards the east". This means horizontal movements away from the subsidence centre (the sinkhole), which is counterintuitive, and should be explained and/or discussed.**

We agree with the reviewer; the observed deformations must be explained in a coherent way, thanks for the inspiring comment. Considering a pure elastic model, the sketch beneath (please see the supplementary file) shows that the evolution of a depression means deformation not only with vertical component but horizontal as well. The farther a point from the center, the more pronounced is the horizontal deformation, which direction points away from the center of the subsidence bowl. Due to the side looking radar geometry, the observed horizontal deformation is not symmetric. However, asymmetry can also be caused by change in the material property (change in geology).

[pls see sketch in supplementary file]

##########

**Reviewer 1, specific comment 4. The equation relating LOS to vertical and horizontal components should also include the heading angle between the track and the north.**

**Reviewer 2, specific comment 4. The equation of decomposition is wrong, the heading angle is missing. please refer to (Fuhrmann & Garthwaite, 2019).**

Thank you for the reviewers to point out this issue. Since the North-South deformations are neglected in the decomposition, the azimuth enters only in the East-West term with the factor, cosine of the heading. Taking into account the heading values for S1, this term is about ±0.96 (depending on pass direction), therefore it was neglected. We acknowledge that equation of decomposition can be misleading in the present form, therefore, we will include the original, full 3D expression in the revised version of the ms. with proper citation.

##########

**Reviewer 1, specific comment 6. Figures 10 shows a cross sections over an area that is highly incoherent, while the lines are continuous from side to side. The authors should explain how this section was made and make it clear where are the true points and where are the lines based on interpolation.**

We agree with the reviewer that it must be emphasized in the ms. that the LoS deformation profiles were constructed by interpolation. We applied a natural neighbor interpolation method based on the points satisfying some distance criteria (< 50 m) around the location of the profile. Therefore, Figs. 10. and 11. show much more the gradual deformation of a zone instead of the single profile. That's the reason we did not mark the location of true points on the deformation curves. This information will be added to the revised ms. Location of cross-sections was selected to get information in roughly perpendicular directions of the area, it was chosen by visual inspection of point distribution.

##########

**Reviewer 1, specific comment 7. Line 183-184: "separation of total line-of-sight (LOS) deformation into east-west and vertical components which can help to understand the mechanism of sinkhole collapse and the progress of underground processes". The authors do not show anywhere later in the paper any insight or discussion regarding the mechanism of sinkhole collapse and progress of underground processes. So what**

is the motivation for this separation to vertical and horizontal components?

The separation of LoS deformation into vertical and horizontal components involves some spatial averaging to resample both datasets to a common grid, which has a smoothing effect. This can help to suppress possible outliers in one hand, on the other hand it is much easier to interpret horizontal and vertical deformations compared to LoS measurements. We agree with the reviewer, that the interpretation was not satisfactory. To constrain the underground processes, we conducted an analytical modeling described at the beginning of authors' responses.

##########

**Reviewer 1, specific comment 8. Lines 257-259: can the authors please explain the large difference between the ascending and descending velocities in the "landslide" area?**

We did not investigate deformation pattern related to the landslide in the original ms., since we focused mining-related displacements, which can help the early identification of sinkhole prone areas.

Based on the relief (represented by SRTM model) and some assumptions about the nature of displacement related to landslide, we can conclude the following. Since this investigation is not part of the ms., the figs referred in the text can be found in the supplementary pdf material.

We take the assumption that the direction of motion is along the local direction of steepest descent, which is shown by the surface gradient vector. This condition is routinely applied not only in landslide mapping applications, but also in investigations related to glacier displacement mapping. The horizontal orientation of the gradient vector in the landslide area varies between -1.8 and -2.2 radian (-103°-126°), where angles are measured relative to the East and increases towards to the North. Therefore, the slope is assumed to move towards south with a slight westward motion (see first fig). The

magnitude of the gradient vector is shown on the second figure, it is about 22-25%, which suggests a moderately steep slope. Assuming that the vertical deformation is more pronounced than the horizontal (westward) motion, which can be plausible based on the gradient vector, the apparent deformation in the LoS direction for the descending pass is larger (see the sketch on the third fig.), that is the reason for the difference. [The incidence angles are 41.4° and 36.3°, for the ascending and descending passes respectively (see Tabl. 2. of the original ms.). ]

[please see figs. in supplementary file]

Orientation angle of SRTM gradient vector (values are in radian relative to East, North is $\pi/2$, South is $-\pi/2$), landslide area is marked by the red box.

Normalized magnitude of the SRTM gradient vector.

##########

**Reviewer 1, specific comment 9. Lines 261-264: The description of the two sides of the bowl is poorly supported by the figure.**

Thanks for the comment, we added Fig. 3. based on source modelling to support the discussion.

##########

**Reviewer 1, specific comments 10. Lines 305-306: "guaranties to maintain coherence" – coherence is definitely not maintained in the central area.**

We agree with the reviewer's remark. It will be emphasized in the revised version of the ms. that the parameters of the Sentinel-1 mission, i.e. short revisit time, orbit positioning control, help to maintain coherence in general; however, any change in the surface backscatter properties can lead to coherence loss.

##########

**Reviewer 2, specific comment 1. The authors talked a lot about geological settings in Section 2 "Geographical and geological background", but it looks not related to your discussions later in the paper. The same problem as in Section 3. Could you relate your deformation results to the geological settings and mining activities? Maybe you can add a section in discussion to talk about the relationship between deformation and geological setting/ mining activities.**

We will add a section devoted to analytical modeling to explain the mechanism of sink-hole formation and its relation to mining activity in the area.

##########

**Reviewer 2, specific comment 5. I think your discussion is not enough, could you please talk about how the deformation results relate to the geological settings you described in Section 2.**

We agree with the reviewer, the results of InSAR analysis is not coupled to the geological setting of the area, thank you for the comment. We will add a source modeling analysis, described above, to find a linkage between deformation evolution and geological setting and explain the subsurface processes.

########## # Reviewer 1, technical corrections 1. The paper requires Language and grammar editing. Lots of sentences lack commas (,) to separate between parts of the sentence. Citation of previous studies should not be in brackets when they are the subject of the sentence. For example, line 143: (Gaidin, 2008) has already drew attention to: : :.., should be: Gaidin (2008) has already: : :.. This type of error appears many times in the paper. 2. Line 115: change horizontal extension to areal extent 3. Line 256: what is MT-InSAR? Fig. 7 is like 6 but descending

**Reviewer 2, technical corrections: 1. Line 307, please check the citation format (Velasco et al., 2017). And some of the same problems across the whole manuscript. 2. Line 221, Small Baseline Subset, SBAS -> Small Baseline Subset (SBAS) 3. Line**

203, 1' resolution SRTM, is it 1 arc second?

Thank you for your comments. We will thoroughly check the English of the ms.

Regarding citations: We used the Mendeley Citation plugin (https://www.mendeley.com/guides/using-citation-editor) for MS Word. It is a rather convenient tool for generating references, citations and bibliographies. We will check and adjust manually the citation format where necessary in the revised ms.

The 1' resolution SRTM was released by USGS, we will add a proper reference in the revised ms, as: Shuttle Radar Topography Mission 1 Arc-Second Global (Digital Object Identifier (DOI) number: /10.5066/F7PR7TFT

MT-InSAR stands for Multi-Temporal InSAR analysis, which is also called TS-InSAR (Time-Series InSAR) in the literature.

[Figure]

Fig. 1. Cumulative LoS deformation computed from the quad-configuration source model (top), deformation from the ascending Sentinel-1 observation (middle) and residuals after subtracting the best-fitting model

[Figure]

**Fig. 1.** Cumulative LoS deformation computed from the quad-configuration source model (top), deformation from the ascending Sentinel-1 observation (middle) and residuals after subtracting the best-fitting mode

[Figure]

Fig. 2. Cumulative LoS deformation of the best-fitting model using four dislocation sources (top), deformation from the descending Sentinel-1 observation (middle) and residuals after subtracting the modeled displacement from the cumulative deformation

**Fig. 2.** Cumulative LoS deformation of the best-fitting model using four dislocation sources (top), deformation from the descending Sentinel-1 observation (middle) and residuals after subtracting the modeled d

[Figure]

Fig. 3. Observed cumulative LoS and best-fitting model LoS deformations along selected profiles (given on Fig. 10. and 11. in the original ms.) for the ascending (top) and descending (bottom) passes.

**Fig. 3.** Observed cumulative LoS and best-fitting model LoS deformations along selected pro-files (given on Fig. 10. and 11. in the original ms.) for the ascending (top) and descending (bottom) passes.

Table 1. Analytical model parameters used in the source modeling. "I" refers to initial, "R" to refined values; coordinates are given in local rectangular system, shown on Fig. 1.

| | model #1 | | model #2 | | model #3 | | model #4 | |
|---|---|---|---|---|---|---|---|---|
| | I | R | I | R | I | R | I | R |
| length [m] | 36 | 24.1 | 72.5 | 63.5 | 80 | 78.2 | 80 | 72.3 |
| width [m] | 42 | 64.0 | 176 | 187.8 | 80 | 81.8 | 90 | 82.1 |
| depth [m] | 180 | 199.7 | 222 | 231.9 | 280 | 273.1 | 295 | 295.9 |
| strike angle [deg] | 5 | 12.8 | 21.5 | 19.1 | 22 | 18.7 | 21 | 17.1 |
| X center [m] | -870 | -880.3 | -1195 | -1259.1 | -1600 | -1630.8 | -1700 | -1700.5 |
| Y center [m] | 1160 | 1195.8 | 956 | 1029.6 | 230 | 224.9 | 810 | 793.1 |
| opening [m] | -3.2 | -2.8 | -1.4 | -1.2 | -3.9 | -2.3 | -3.8 | -3.1 |

**Fig. 4.** Table 1. Analytical model parameters used in the source modeling. "I" refers to initial, "R" to refined values; coordinates are given in local rectangular system, shown on Fig. 1.

---

## Referee Report (RR1)

New comments on the revised version

Lines 83-87: How is the pollution problem related to this paper?

Figure 2: draw a rectangle around the study area.

Line 140: change horizontal extension to surface area

Fig. 3a legend: correct: burried > buried; profil > profile; residental > residential; Fig. 3d correct: estabilishments > establishments.

Line 156: and beginning of the 1990s

Table 1: correct impLOSion > implosion (or collapse)

Line 229: "…which can help to understand the mechanism of sinkhole collapse and the progress of underground processes." Where is that shown later?

Line 231: winter scenes with snow cover were excluded from the analysis: how does this affect time series?

Line 277: "15 m x 15 m pixel size in the slant range" – should be in range and azimuth (or simply 15 m x 15 m pixel size).

Line 310 and Figure 4: Again, the heading angle is ignored (or at least explained).

Line 339: "on the left bank of the Tisza river" – left or right depend on where you come from. Use east, west, north, south.

Figures 5 and 6 show 5 points each, and figures 7 and 8 show their deformation series. It would be useful and insightful if the authors show the same points (or at least the closest ones) from the ascending and descending tracks. The way shown here makes it difficult to follow and to judge how consistent and reliable the results are. Also, the order of the points in the legend is arbitrary and different in the two figures, adding to the difficulty in following the results. This adds to a related question I asked in the previous review about comparison between the ascending and descending results.

Line 344: "along a cross-section directed almost north-south (cross-section A-B)" – please add the cross section to figure 5 to let the reader judge between actual points and interpolation intervals. (same for the E-W cross section). I also suggest (as I suggested before) that the plots need different markings for actual and interpolated points.

My specific comments 3 and 5 did not get an appropriate reply. Most of the questions in these comments were ignored. The graph showing the expected horizontal components is incorrect for a case where the central part is a void towards which all the material collapses. Thus, the question of the counterintuitive direction of horizontal movements still requires explanation.

Lines 356-358: these lines talk about the landslide in the north and are not related to the sentence before or after them. I also asked a question about the difference between the ascending and descending results of this area and do not see an answer in the text.

Lines 360-363: The readers should have the ability to judge for themselves if the steps and gradual parts of the cross sections are real or are due to lack in data points. See previous comment (line 344).

Lines 455-456: "The observed gradual subsidence also supports the assumption of pure elastic deformation" – this should be explained, as longer periods and gradual deformation support plastic (or viscous), rather than elastic deformation. In lines 582-584 the authors cite a paper that "confirmed that the main driving mechanism of sinkhole formation in the area is much more like the mechanism of the perfect suffosion of non-cohesive soils, rather than the sudden dropout of cohesive soils". Therefore, I think that elastic behaviour is highly questionable here. There are two recent papers that address this problem which the authors should look at: Atzori et al., 2015, Geophys. Res. Lett., 42, and Baer et al., 2018, J. Geophys. Res. Earth Surf. 123, 678–693.

Lines 510-530: please show the location and traces of the 4 models and the actual mines on the maps of Figures 13 and 14 and on the cross sections (Fig. 15). It can be useful to show the actual mines also on the deformation maps (Figs 5, 6).

---

## Author Response (AR2)

We thank the Editor and Reviewer#2 for these new comments.

Editorial comment:

The paper needs to be restructured before publication. The discussion part of the paper is too short and is focused only on monitoring aspect of the work. There is no discussion about modeling results. The discussion about source modeling needs to be shifted from the result section (page 12, 13) to the discussion section and to be elaborated more there. The same problem exists for the conclusion, which is focused only on monitoring aspect of the work.

Thank you for your comment, the ms. was restructured as suggested.

The InSAR profiles plotted in Fig. 15 are continuous. I assume that the authors have done interpolation between InSAR points falling into the profiles plotted in Figs. 9-10. Please use original InSAR observations with their corresponding error bars, and not interpolated ones, for comparison with modeling. Due to a large InSAR data gap, in particular along profile AB, interpolation fails to represent the true behavior of the real data here, which may contribute to the large discrepancy that is observed currently between modeling and AB profile in Fig. 15. Please address this in the revised version

Thank you for your remark. All the figures showing the cross-section of the deforming area were completed with the location of point data to better discriminate raw data from interpolated values. This was also suggested by Reviewer#2.

Report #2 - New comments on the revised version

Lines 83-87: How is the pollution problem related to this paper?

In the first round of the review process a clear motivation of the paper was missed and the significance of the problem was lacked by the reviewers. With this analogy we wanted to emphasize the severity of the situation. Sinkhole development causes local problems in several parts of the world, it is a well-known problem along the Dead Sea. However, in our case the situation is different and brine pollution due to sinkhole collapse can evolve to a regional massive ecological problem similar to the metal pollution occured a few years ago. The metal pollution was mentioned in this context.

Figure 2: draw a rectangle around the study area.

The study area, Solotvyno, is already highlighted in red.

Line 140: change horizontal extension to surface area

Thanks, it was corrected.

Fig. 3a legend: correct: burried > buried; profil > profile; residental > residential; Fig. 3d correct: estabilishments > establishments.

Thanks, typos were corrected.

Line 156: and beginning of the 1990s

Thanks, it was corrected.

Table 1: correct impLOSion > implosion (or collapse)

Thanks, it was corrected.

Line 229: "…which can help to understand the mechanism of sinkhole collapse and the progress of underground processes." Where is that shown later?

+ My specific comments 3 and 5 did not get an appropriate reply. Most of the questions in these comments were ignored. The graph showing the expected horizontal components is incorrect for a case where the central part is a void towards which all the material collapses. Thus, the question of the counterintuitive direction of horizontal movements still requires explanation.

The sentence in question was modified. The modelling showed that it is not straightforward to interpret the observed deformations as we face with several sources and not just one void in the centre of the area.

Line 231: winter scenes with snow cover were excluded from the analysis: how does this affect time series?

Significant change, like snow, in the backscattering properties of a resolution element can cause signal decorrelation. Leaving out snowy scenes results in no measurement for the epoch of the specific acquisition. For the largest average velocity (5 cm/yr) with only S1A observations (12 days), the expected deformation change is 0.167 cm. So leaving out a few scenes will not cause signal aliasing problems.

Line 277: "15 m x 15 m pixel size in the slant range" – should be in range and azimuth (or simply 15 m x 15 m pixel size).

Thanks for the remark, it was corrected.

Line 310 and Figure 4: Again, the heading angle is ignored (or at least explained).

The heading angle is not ignored in the original expression of the line of sight deformation decomposition given in l. 200 in the revised ms. For a north-south orbiting satellite the cosine of the heading angle is appr. -1, while the sine is near zero. The cosine term enters in the east-west deformation component, while the sine in the north-south. The problem is underdetermined (from 2D to 3D) therefore, the north-south deformation component is usually neglected as shown by equation given in l. 206. of the revised ms.

Line 339: "on the left bank of the Tisza river" – left or right depend on where you come from. Use east, west, north, south.

The left/right bank of a river is determined by its flow direction.

Figures 5 and 6 show 5 points each, and figures 7 and 8 show their deformation series. It would be useful and insightful if the authors show the same points (or at least the closest ones) from the ascending and descending tracks. The way shown here makes it difficult to follow and to judge how consistent and reliable the results are. Also, the order of the points in the legend is arbitrary and different in the two figures, adding to the difficulty in following the results. This adds to a related question I asked in the previous review about comparison between the ascending and descending results.

"The analysis of the previous section overlooks a fundamental fact: different acquisition geometries see, with a few exceptions, different objects and each InSAR data-stack has dissimilar families of measurement points." (A. Ferretti: Satellite InSAR data, EAGE publications, 2014, p. 92.)

Based on the above, the direct comparison of ascending and descending deformation series is not adequate for natural scatterers. However, artificial twin scatterers with well-known location of the stable phase centre allows collocated measurements. It is especially true for our case, where distributed scatterers are more dominant. In the revised ms. we implied to this fact and that's the reason we did not try to compare time series of individual scatterers of ascending and descending tracks. The colorscale for Figs. 5 and 6 are the same which can support some visual comparison, but we would rather not compare measurement points of different scattering mechanism.

To remedy this shortcoming, we followed Ferretti (2014) in our analysis.
" To overcome this limitation the area of interest is divided into small patches of terrain. All measurement points belonging to the same patch and to the same acquisition geometry are averaged …"
As described in sec. 4.2 in the revised ms.

Line 344: "along a cross-section directed almost north-south (cross-section A-B)" – please add the cross section to figure 5 to let the reader judge between actual points and interpolation intervals. (same for the E-W cross section). I also suggest (as I suggested before) that the plots need different markings for actual and interpolated points.

Figs. 5. and 6. are already full with symbols and you pointed out in the above comment, that's the reason we decided to mark the location of the cross-sections on Figs. 9 and 10. (please see the insets) in the revised ms. We agree with you to discriminate points which deformation history is the result of interpolation from measurement points. We marked the latter ones with blue rectangles on the cross-section plots.

Lines 356-358: these lines talk about the landslide in the north and are not related to the sentence before or after them. I also asked a question about the difference between the ascending and descending results of this area and do not see an answer in the text.

Since the landslide is not related to the deformation of the mining territory we replied your question not in the ms., but in the separate file, named "Auhtor's response". The answer can be found on p. 9-11.

Lines 360-363: The readers should have the ability to judge for themselves if the steps and gradual parts of the cross sections are real or are due to lack in data points. See previous comment (line 344).

Yes, we agree with you and marked the data points on the cross-section plots so the readership of the journal can clearly discriminate data points from interpolation points.

Lines 455-456: "The observed gradual subsidence also supports the assumption of pure elastic deformation" – this should be explained, as longer periods and gradual deformation support plastic (or viscous), rather than elastic deformation. In lines 582-584 the authors cite a paper that "confirmed that the main driving mechanism of sinkhole formation in the area is much more like the mechanism of the perfect suffosion of non-cohesive soils, rather than the sudden dropout of cohesive soils". Therefore, I think that elastic behaviour is highly questionable here. There are two recent papers that address this problem which the authors should look at: Atzori et al., 2015, Geophys. Res. Lett., 42, and Baer et al., 2018, J. Geophys. Res. Earth Surf. 123, 678–693.

Thank you for pointing out these two papers describing sinkhole modelling. Atzori et al. (2015) investigates sinkholes along the Dead Sea shorelines and utilizes purely elastic inverse modelling to constrain contemporary evolution of surface deformation. We also utilized elastic modelling in our computation, which can be easily implemented and provides a good first approximation of the surface deformation. It is especially valid in the early stage of sinkhole evolution as the investigations of Atzori et al. (2015) also confirms.  The numerical simulations conducted by Baer et al. (2018) with different viscosity parameters definitely ensure a more complex interpretation of sinkhole related subsidence. We will study it thoroughly in the future.

Lines 510-530: please show the location and traces of the 4 models and the actual mines on the maps of Figures 13 and 14 and on the cross sections (Fig. 15). It can be useful to show the actual mines also on the deformation maps (Figs 5, 6).

Thank you for the remark, the boundary of mines (grey polygons) was added to the figures showing source modelling results as well as the location of the source models (black dots).

[revised manuscript text omitted]

---

## Author Response (AR3)

Dear Prof. Motagh,

thank you for your comments and suggestions.

Minor comment:
page 2 : Fig.1 illustrates the most recent deformation
I would change: Fig. 1 illustrates the most recent sinkhole developed in the area in 2012

Thank you for your remark, the text has been changed.

Page 3: The Solostuyno salt diapir.......
please polish the sentence. It is not grammatically correct (In general, I would recommend polishing the manuscript by a native speaker or a professional service)

Thank you for your comment, the sentence was rephrased.

Material and Methods
In this section we only see description about InSAR analysis. Later in the Result section (section 4), we again see switching to methodology for the source moldeling. I would recommend bring the theoretical aspect of the modeling to the Material and Methods to have a coherent result section (section 4) before Discussion.

Thank you for your comment, the theoretical description of the modeling was moved to section 3.

Page 7. Time-series of Sentinel-1 ....... deformation beneath the city of Solotvyno
Do you mean south of Solotvyno? The deformation zone seems to be outside the city. Anyway, please indicate the location of the city in Fig. 4 for a better clarification

Fig. 4. shows the concept of LOS decomposition. Fig 5. depicts the LOS deformation. The background of Fig. 5. is a Google Earth satellite image, the zoom level was set to fully match the extension of the city, that is the reason we did not mark the boundary of Solotvyno. The mining area is situated in the center of the city. The river on the south and east is the natural boundary of Solotvyno (and it is the border between Ukraine and Romania).